# COMPRESSING LATENT SPACE VIA LEAST VOLUME

**Qiuyi Chen & Mark Fuge**
Department of Mechanical Engineering
University of Maryland, College Park
`{qchen88,fuge}@umd.edu`

## ABSTRACT

This paper introduces *Least Volume*—a simple yet effective regularization inspired by geometric intuition—that can reduce the necessary number of latent dimensions needed by an autoencoder without requiring any prior knowledge of the intrinsic dimensionality of the dataset. We show that the Lipschitz continuity of the decoder is the key to making it work, provide a proof that PCA is just a linear special case of it, and reveal that it has a similar PCA-like importance ordering effect when applied to nonlinear models. We demonstrate the intuition behind the regularization on some pedagogical toy problems, and its effectiveness on several benchmark problems, including MNIST, CIFAR-10 and CelebA.

## 1 INTRODUCTION

Learning data representation is crucial to machine learning (Bengio et al., 2013). On one hand, a good representation can distill the primary features from the data samples, thus enhancing downstream tasks such as classification (Krizhevsky et al., 2017; Simonyan & Zisserman, 2014; He et al., 2016). On the other hand, when the data representation lies in a low dimensional latent space $Z$ and can be mapped backward to the data samples via some decoder $g$, we can considerably facilitate generative tasks by training generative models in $Z$ (Ramesh et al., 2021; Rombach et al., 2022).

But what makes a data representation—*i.e.*, $\mathcal{Z} = e(\mathcal{X}) \subset Z$ of a dataset $\mathcal{X}$—good? Often, a low dimensional $Z$ is preferred. It is frequently hypothesized that a real world dataset $\mathcal{X}$ in high dimensional data space $X$ only resides on a low dimensional manifold (Fefferman et al., 2016) (or at least most part of $\mathcal{X}$ is locally Euclidean of low dimension), hence due to the rank theorem (Lee, 2012), $\mathcal{X}$'s low dimensionality will be inherited by its latent set $\mathcal{Z}$ through an at least piecewise smooth and constant rank encoder $e$ (*i.e.*, $\mathcal{Z}$ is low dimensional even if $Z$ is high dimensional).

Therefore, for a $\mathcal{Z} \subset Z$ retaining sufficient information about $\mathcal{X}$, a low dimensional $Z$ can provide several advantages. First, it can improve the efficacy of downstream tasks by aligning its latent dimensions more with the informative dimensions of $\mathcal{Z}$ and alleviating the curse of dimensionality. In addition, it can increase the robustness of tasks such as data generation. Specifically, if a subset $U \subseteq \mathcal{Z}$ constitutes an $n$-D manifold that can be embedded in an $n$-D Euclidean space, then due to the invariance of domain (Bredon, 2013), $U$ is *open* in $Z$ as long as $Z$ is $n$-D. Thanks to the basis criterion (Lee, 2010), this openness means for a conditional generative model trained in such $Z$, if its one prediction $\hat{z}$ is not far away from its target $z \in U$, then $\hat{z}$ will also fall inside $U \subseteq \mathcal{Z}$, thus still be mapped back into $\mathcal{X}$ by $g$ rather than falling outside it. Moreover, in the case where $\mathcal{Z}$ is a manifold that cannot embed in similar dimensional Euclidean space, or where $\mathcal{Z}$ is not even a manifold, retrieving the least dimensional $Z$ needed to embed $\mathcal{Z}$ may pave the way for studying the complexity and topology of $\mathcal{Z}$ and $\mathcal{X}$. Additionally, people also hope the latent representation can indicate the amount of information each dimension has (Rippel et al., 2014; Pham et al., 2022), so that the data variations along the principal dimensions can be easily studied, and trade-offs can be easily made when it is necessary to strip off less informant latent dimensions due to computational cost. This is why PCA (Pearson, 1901), despite being a century old, is still widely applied to different areas, even if it is a simple linear model.

To automatically learn a *both* low dimensional and ordered nonlinear latent representation, in this work we introduce the Least Volume regularization, which is based on the intuition that packing a flat paper into a box consumes much less space than a curved one. This paper's contributions are:

1. We introduce *Least Volume* (LV) regularization for autoencoders (AEs) that can compress the latent set into a low dimensional latent subspace spanned by the latent space's standard basis. We show that upper bounding the decoder's Lipschitz constant is the key to making it work, and verify its necessity with an ablation study.

2. We prove that PCA is exactly a special case of least volume's application on a linear autoencoder. In addition, we show that just like in PCA, there is a close-to-1 positive correlation between the latent dimension's degree of importance and standard deviation.

3. We apply least volume to several benchmark problems including a synthetic dataset with known dimensionality, MNIST and CIFAR-10, and show that the volume penalty is more effective than the traditional regularizer Lasso in terms of compressing the latent set. We make the code public on GitHub[1] to ensure reproducibility.

## 2 METHODOLOGY AND INTUITION

As proved later in Theorem 2, if a continuous autoencoder can reconstruct the dataset perfectly, then its latent set is a homeomorphic copy of it. Concretely speaking, if through the lens of the manifold hypothesis we conceive the dataset as an *elastic* curved surface in the high dimensional data space, then the latent set can be regarded as an intact "flattened" version of it tucked into the low dimensional latent space by the encoder. Therefore, the task of finding the least dimensional latent space can be imagined as the continuation of this flattening process, in which we keep compressing this elastic latent surface onto latent hyperplanes of even lower dimensionality until it cannot be flattened anymore. Thereafter, we can extract the final hyperplane as the desired least dimensional latent space. Ideally, we prefer a hyperplane that is either perpendicular or parallel to each latent coordinate axis, such that we can extract it with ease.

### 2.1 VOLUME PENALTY

To flatten the latent set and align it with the latent coordinate axes, we want the latent code's standard deviation (STD) vector $\boldsymbol{\sigma}$ to be as sparse as possible, which bespeaks the compression of the dataset onto a latent subspace of the least dimension. The common penalty for promoting sparsity is the $L_1$ norm. However, $\|\boldsymbol{\sigma}\|_1$ does not necessarily lead to flattening, as we will discuss later in §A.1.1.

An alternative regularizer is $\prod_i \sigma_i$—the *product of all elements of the latent code's STD vector $\boldsymbol{\sigma}$*. We call this the **volume** penalty. It is based on the intuition that a curved surface can only be enclosed by a cuboid of much larger volume than a cuboid that encloses its flattened counterpart. The cuboid has its sides parallel to the latent coordinate axes (as shown in Fig. 1) so that when its volume is minimized, the flattened latent set inside is also aligned with these axes. To evaluate the cuboid's volume, we can regard the STD of each latent dimension as the length of each side of this cuboid. The cuboid's volume reaches zero only when one of its sides has zero length (*i.e.*, $\sigma_i = 0$), indicating that the latent surface is compressed into a linear subspace. Conceptually, we can then extract this subspace as the new latent space and continue performing this flattening process *recursively* until the latent set cannot be flattened any more in the final latent space.

In practice, though, this recursive process is troublesome to implement, whereas directly minimizing the volume may induce vanishing gradients when several $\sigma_i$ are marginal, hindering flattening. Therefore, we supplement each $\sigma_i$ with an amount $\eta \geq 0$ to make the volume penalty become $\prod_i(\sigma_i + \eta)$. To avoid extremely large gradients when the latent dimension $m$ is large, we implement instead equivalently minimizing the geometric mean $\sqrt[m]{\prod_i(\sigma_i + \eta)}$, which we evaluate using the ExpMeanLog trick to avoid numerical issues.

### 2.2 LIPSCHITZ CONTINUITY REGULARIZATION ON DECODER

Mechanically encouraging such latent sparsity, however, can lead to a trivial solution where we drive all latent STDs close to zero without learning anything useful. This can occur when the elastic latent surface shrinks itself *isotropically* to naïvely shorten all of the enclosing cuboid's sides, without

---

[1]`https://github.com/IDEALLab/Least_Volume_ICLR2024`

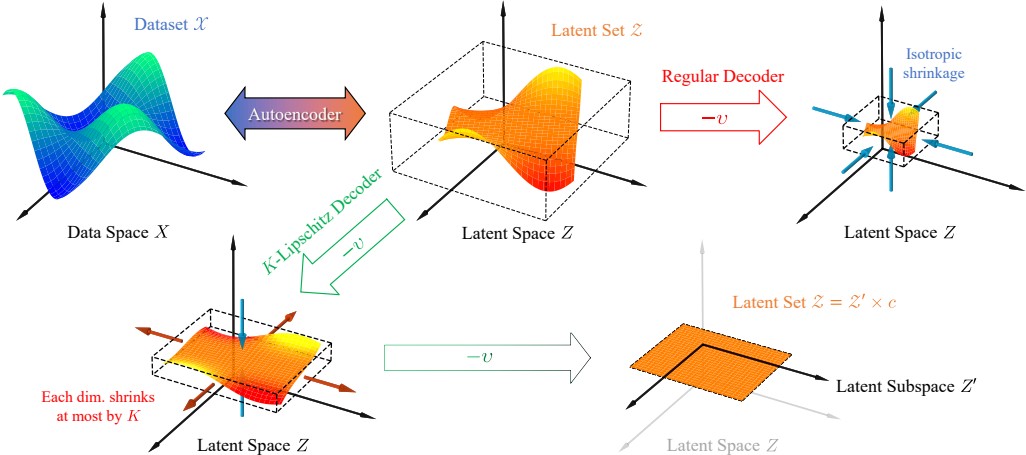

Figure 1: Flattening the latent set via Least Volume ("$-v$" means reducing the cuboid's volume).

further flattening the latent surface. To forestall this arbitrary shrinking that causes the trivial solution, we need to properly regularize the autoencoder's latent set's elasticity.

The appropriate regularization turns out to be the *Lipschitz continuity of the decoder*. The intuition is as follows. In the trivial solution, the encoder would naïvely compresses all latent STDs to almost zero. The decoder would then fight the encoder's shrinking by scaling up its network weights such that the now small perturbations in the latent input still induce large enough output variations in data space to achieve low reconstruction error. To do this, however, the decoder must possess a large Lipschitz constant ($K$) by definition. If instead, the decoder had a small bounded Lipschitz constant, then any latent dimension with STD close to zero must also have close-to-zero variation in the data space due to the small $K$ and thus correspond to a dimension perpendicular to the data surface. Meanwhile, a principal data dimension must retain a large variation in the latent space, as otherwise, the decoder would need to violate its bounded Lipschitz constant to achieve low reconstruction error. This bounded Lipschitz constant on the decoder thereby prevents the encoder from arbitrarily collapsing the latent codes to the trivial solution. Figure 1 illustrates this intuition. Ghosh et al. (2019) imposed Lipschitz constraint on the decoder under a similar motivation.

To achieve this regularization, Lipschitz continuity can be conveniently hard-constrained via *spectral normalization* (Miyato et al., 2018), namely normalizing the spectral norm (the maximum singular value) of all linear layers to 1. With 1-Lipschitz functions such as LeakyReLU as activation functions, the Lipschitz constant of the decoder is guaranteed to be not greater than 1. For accurate regularization of any convolutional layers, we employ the power method in (Gouk et al., 2021).

## 2.3 LEAST VOLUME FORMULATION

In summary, for an unsupervised representation learning problem in which the latent set $\mathcal{Z} = e_\theta(\mathcal{X})$ is required to preserve enough information of a dataset $\mathcal{X}$—as per the criterion that it minimizes a reconstruction loss function $J(g_\theta(\mathcal{Z}), \mathcal{X})$, conceptually the *Least Volume* (LV) problem is:

$$\arg\min_\theta \quad L_{\text{vol}}(\mathcal{Z}) = L_{\text{vol}}\big(e_\theta(\mathcal{X})\big) \tag{1}$$

$$\text{s.t.} \quad \mathcal{Z} \in \mathcal{Z}^\star := \{\mathcal{Z} = e_\theta(\mathcal{X}) \mid \theta \text{ minimizes } J(g_\theta \circ e_\theta(\mathcal{X}), \mathcal{X})\} \tag{2}$$

$$\|g_\theta(z_1) - g_\theta(z_2)\| \leq K\|z_1 - z_2\|, \quad \forall\{z_1, z_2\} \subseteq Z \tag{3}$$

Observe that for any homeomorphism $h$—a bijective continuous function whose inverse is also continuous—the new latent set $h(\mathcal{Z}) = (h \circ e_\theta)(\mathcal{X})$, which is homeomorphic to $\mathcal{Z}$, is equivalent to $\mathcal{Z}$ in the sense that $g_\theta \circ e_\theta(x) = (g_\theta \circ h^{-1}) \circ (h \circ e_\theta)(x)$. Therefore, as long as $g_\theta$ and $e_\theta$ have enough complexity to respectively represent $g_\theta \circ h^{-1}$ and $h \circ e_\theta$ for a given set $\mathcal{H}$ of $h$, each homeomorphic latent set $h(\mathcal{Z})$ with $h \in \mathcal{H}$ must also be an optimal solution to $J$, thus residing in $\mathcal{Z}^\star$. Hence, the more complexity $g_\theta$ and $e_\theta$ have, the larger the set $\mathcal{H}$ is, thus a more complicated homeomorphism $h$

we can obtain to flatten $\mathcal{Z}$ more. In reality, though, due to the complexity of this optimization, we need to resort to the weighted sum objective

$$L = J + \lambda \cdot L_{\text{vol}} \tag{4}$$

to derive an approximate solution, where $\lambda$ needs to be fine-tuned.

## 3 THEORETICAL ANALYSIS

Given the least volume formulation, in this section we formalize its surface-flattening intuition with theoretical analysis, inspect the introduced variables' properties and their relationships, and demonstrate its equivalence to PCA in the linear special case, before moving onto our empirical results in §5. For space reasons, many of the proof details are placed in §A of the appendix.

### 3.1 WHAT EXACTLY DO WE MEAN BY VOLUME?

The volume $\prod_i \sigma_i$ of a latent set is the square root of the diagonal product of the covariance matrix $S$ of the latent codes. Apart from this trivial fact, below we show that its square is the tight upper bound of the latent determinant $\det S$, which is referred to as *Generalized Variance* (GV) in some literature.

**Theorem 1.** *The square of volume—i.e., $\prod_i \sigma_i^2$ —is a tight upper bound of the latent determinant $\det S$. If $\det S$ is lower bounded by a positive value, then $\prod_i \sigma_i^2$ is minimized down to $\det S$ if and only if $S$ is diagonal.*

Theorem 1 can lead to some interesting results that we will see in §3.4. Its proof in given in §A.1. Here the *positive lower bound* of $\det S$ is an unknown inherent constant determined by the dataset, and it originates from the intuition that when the latent space reduces into the least dimensional one in which $\mathcal{Z}$ cannot be flattened anymore, then $\det S$ cannot be zero, as otherwise it suggests $\mathcal{Z}$ resides in a linear latent subspace and creates contradiction. Nor should $\det S$ be arbitrarily close to 0, as the $K$-Lipschitz decoder prevents degenerate shrinkage.

As mentioned in §2.1, in practice we instead minimize the supplemented geometric mean $\sqrt[m]{\prod_i (\sigma_i + \eta)}$ with $\eta \geq 0$ to avoid the recursive subspace extraction process in conception. Minimizing this variable not only equivalently minimizes an upper bound of the volume, but also naturally interpolates between $\sqrt[m]{\prod_i \sigma_i}$ and $\frac{1}{m}\|\boldsymbol{\sigma}\|_1$ gradient-wise, given that as $\eta \to \infty$:

$$\nabla_\theta \sqrt[m]{\prod_i (\sigma_i + \eta)} = \frac{1}{m} \sum_i \frac{\sqrt[m]{\prod_i (\sigma_i + \eta)}}{\sigma_i + \eta} \cdot \nabla_\theta \sigma_i \quad \longrightarrow \quad \frac{1}{m} \sum_i \nabla_\theta \sigma_i = \nabla_\theta \frac{1}{m} \|\boldsymbol{\sigma}\|_1 \tag{5}$$

So we can seamlessly shift from volume penalty to $L_1$ penalty by increasing $\eta$. We can see that the lower the $\eta$, the more the regularizer's gradient prioritizes minimizing smaller $\sigma_i$, which intuitively should make $\boldsymbol{\sigma}$ sparser than the $L_1$ penalty does that scales every $\nabla_\theta \sigma_i$ equally. We shall see in Section 5 that $\sqrt[m]{\prod_i (\sigma_i + \eta)}$ is indeed more efficient. For a further simple pedagogical example where minimizing the volume produces sparser and more compact latent sets than the $L_1$ norm, we refer readers to §A.1.1 in the appendix. More relevant discussions can be found in §C.

### 3.2 AN ERRORLESS CONTINUOUS AUTOENCODER LEARNS A TOPOLOGICAL EMBEDDING

It is pointless to only minimize the volume, given that we can always reduce it to 0 by trivially encoding every data sample to the same latent point. To make the "flattening" process in §2 meaningful, we need to ensure the autoencoder has good reconstruction performance over the dataset, such that the latent set is a low dimensional replica of the dataset that preserves useful topological information. This is justified by the following propositions. Their proofs are given in §A.2.

**Lemma 1.** *If $\forall x \in \mathcal{X}$, $g \circ e(x) = x$, then both $e$ and $g$ are bijective between $\mathcal{X}$ and $\mathcal{Z} = e(\mathcal{X})$.*

Although some regard an errorless autoencoder as "trivial" since it only learns an one-to-one representation, it is actually not as trivial as they think, thanks to the *continuity* of the autoencoder.

**Theorem 2.** *A continuous autoencoder between the data space $X = \mathbb{R}^n$ and the latent space $Z = \mathbb{R}^m$ with the norm-based reconstruction error $\|g \circ e(x) - x\| = 0$ everywhere on the dataset $\mathcal{X} \subseteq X$ learns a topological embedding of the dataset. In other words, the latent set $\mathcal{Z} = e(\mathcal{X}) \subseteq Z$ is a homeomorphic copy of the dataset.*

Because $\mathcal{Z}$ is a homeomorphic copy of $\mathcal{X}$, $\mathcal{X}$'s important topological properties like connectedness and compactness—which are invariant under homeomorphism—are preserved on $\mathcal{Z}$. This means if some of these invariant properties are the only information we rely on when analyzing $\mathcal{X}$, then analyzing $\mathcal{Z}$ is equivalent to directly analyzing $\mathcal{X}$. For instance, if we want to know how many disconnected components $\mathcal{X}$ has, or evaluate its local dimensionality, then theoretically we can derive the same results from $\mathcal{Z}$. Of course, $\mathcal{Z}$ becomes more efficient to analyze if it resides in a low dimensional linear subspace easy to extract.

**Corollary 2.1.** *If the above errorless autoencoder's latent set has the form $\mathcal{Z} = \mathcal{Z}' \times c$ where $c \in \mathbb{R}^p$ is constant and $\mathcal{Z}' \subseteq Z' = \mathbb{R}^{m-p}$, then $\pi \circ e|_\mathcal{X}$ is also a topological embedding of $\mathcal{X}$, where $\pi : Z' \times c \to Z'$ is the projection map.*

Corollary 2.1 suggests that if we can force the errorless autoencoder's latent set to assume such a form while increasing the vector $c$'s dimension $p$ as much as possible, we can obtain a latent space that is the lowest dimensional Euclidean space that can still correctly embed the dataset. If achieved, the resulting low dimensional latent space is not only more efficient to analyze, but also provides useful information about the topology of $\mathcal{X}$. For instance, not every smooth $m$ dimensional manifold can be embedded in $\mathbb{R}^m$ (*e.g.*, sphere and Klein bottle), but the Whitney Embedding Theorem (Lee, 2012) states that it can be embedded in $\mathbb{R}^{2m}$. Thus if $\mathcal{X}$ is a smooth manifold, then obtaining the least dimensional $Z$ through a *smooth* autoencoder can provide a lower bound of $\mathcal{X}$'s dimensionality.

In practice, the autoencoder's ideal everywhere-errorless-reconstruction is enforced by minimizing the mean reconstruction error $\epsilon = \mathbb{E}\|g \circ e(x) - x\|$, yet due to inevitable numerical errors and data noise that should be ignored by the autoencoder reconstruction, $\epsilon$ cannot strictly reach zero after optimization. We can only empirically set a pragmatic tolerance $\delta$ for it, and regard the autoencoder as having approximately learned the topological embedding when $\epsilon < \delta$. Likewise, numerically the latent set cannot strictly take the form in Corollary 2.1, but rather only at best has marginal STDs in several latent dimensions. Intuitively, we may consider the latent set as flattened into a thin plane and discard these nearly-constant latent dimensions to obtain a latent set of lower dimensionality with $\pi$. But how do we properly trim off these dimensions? And more importantly, is it safe?

## 3.3 Is it Safe to Prune Trivial Latent Dimensions?

After training an autoencoder with least volume, we can expect several latent dimensions learned by the encoder $e$ to have STDs close to zero. Although such latent set $\mathcal{Z}$ is not exactly in the form of $\mathcal{Z}' \times c$ in Corollary 2.1 for us to apply $\pi$, we can still regarded its dimensions of marginal STDs as trivial latent dimensions, in the sense that *pruning* them—*i.e.*, fixing their value at their mean—will not induce much difference to the decoder $g$'s reconstruction, provided that $g$'s Lipschitz constant is not large. This is justified by the following theorem, whose proof is presented in §A.3.

**Theorem 3.** *Suppose the STD of latent dimension $i$ is $\sigma_i$. If we fix $z_i$ at its mean $\bar{z}_i$ for each $i \in P$ where $P$ is the set of indices of the dimensions we decide to prune, then the $L_2$ reconstruction error of the autoencoder with a $K$-Lipschitz decoder $g$ increases by at most $K\sqrt{\sum_{i \in P} \sigma_i^2}$.*

The pruned $\tilde{\mathcal{Z}}$ is then of the very form $\tilde{\mathcal{Z}} = \mathcal{Z}' \times c$ and can be fed to $\pi$ to extract $\mathcal{Z}'$. The inverse $\pi^{-1}$ helps map $\mathcal{Z}'$ back into the data space through $g \circ \pi^{-1}$ without inducing large reconstruction error.

This theorem also supports our intuition about why having a Lipschitz continuous decoder is necessary for learning a compressed latent subspace—for small $K$, a latent dimension of near-zero STD cannot correspond to a principal dimension of the data manifold. In contrast, in the absence of this constraint, the decoder may learn to scale up $K$ to align near-zero variance latent dimensions with some of the principal dimensions.

## 3.4 Relationship to PCA and the Ordering Effect

Surprisingly, as shown in §A.4, one can prove that PCA is a linear special case of least volume. This implies there could be more to volume minimization than just flattening the surface.

**Proposition 1.** *An autoencoder recovers the principal components of a dataset $X$ after minimizing the volume, if we:*

1. *Make the encoder $e(x) = Ax + a$ and the decoder $g(z) = Bz + b$ linear maps,*

2. *Force the reconstruction error $\|g \circ e(X) - X\|_F$ to be strictly minimized,*

3. *Set the Lipschitz constant to 1 for the decoder's regularization,*

4. *Assume the rank of the data covariance $\mathrm{Cov}(X)$ is not less than the latent space dimension $m$.*

*Specifically, $B$'s columns consist of the principal components of $X$, while $A$ acts exactly the same as $B^\top$ on $X$. When $X$ has full row rank, $A = B^\top$.*

The proof of Proposition 1 in §A.4 suggests that the ordering effect of PCA—*i.e.*, the magnitude of its each latent dimension's variance indicates each latent dimension's degree of importance—is at least partially a result of reducing the latent determinant until the singular values of $B$ reach their upper bound 1, such that $g$ becomes isometric and *preserves distance*. In other words, this means $e$ is not allowed to scale up the dataset $X$ along any direction in the latent space if it is unnecessary, as it will increase the latent determinant, and thus the volume. Therefore, likewise, although Theorem 3 does not prevent a latent dimension of large STD from being aligned to a trivial data dimension, we may still expect minimizing the volume or any other sparsity penalties to hinder this unnecessary scaling by the encoder $e$, given that it increases the penalty's value. This suggests that the latent dimension's importance in the data space should in general scale with its latent STD.

To investigate if least volume indeed induces a similar importance ordering effect for nonlinear autoencoders, we need to first quantify the importance of each latent dimension. This can be done by generalizing the *Explained Variance* used for PCA, after noticing that it is essentially *Explained Reconstruction*:

**Proposition 2.** *Let $\lambda_i$ be the eigenvalues of the data covariance matrix. The explained variance $\mathcal{R}(\{i\}) = \frac{\lambda_i}{\sum_j^n \lambda_j}$ of a given latent dimension $i$ of a linear PCA model is the ratio between $\mathcal{E}_{\|\cdot\|_2^2}(\{i\})$—the MSE reconstruction error induced by pruning this dimension (i.e., setting its value to its mean 0) and $\mathcal{E}_{\|\cdot\|_2^2}(\Omega)$—the one induced by pruning all latent dimensions, where $\Omega$ denotes the set of all latent dimension indices.*

So for PCA, the explained variance actually measures the contribution of each latent dimension in minimizing the MSE reconstruction error $\mathbb{E}\|x - \hat{x}\|_2^2$ as a percentage. Since for a nonlinear model the identity $\mathcal{R}(\{i\}) + \mathcal{R}(\{j\}) = \mathcal{R}(\{i, j\})$ generally does not hold, there is no good reason to stick with the MSE. It is then natural to extrapolate $\mathcal{R}$ and $\mathcal{E}$ to our nonlinear case by generalizing $\mathcal{E}_{\|\cdot\|_2^2}(P)$ to $\mathcal{E}_D(P)$, *i.e.*, the *induced reconstruction error w.r.t. metric $D$ after pruning dimensions in $P$*:

$$\mathcal{E}_D(P) = \mathbb{E}[D(\tilde{x}_P, x)] - \epsilon = \mathbb{E}[D(\tilde{x}_P, x)] - \mathbb{E}[D(\hat{x}, x)] \tag{6}$$

Then $\mathcal{R}_D(P)$—the *explained reconstruction* of latent dimensions in $P$ w.r.t. metric $D$—can be defined as $\mathcal{R}_D(P) = \mathcal{E}_D(P)/\mathcal{E}_D(\Omega)$. More details about this extrapolation can be found in Remark 2.1.

For our experiments, we choose $L_2$ distance as $D$, *i.e.*, $L_2(\tilde{x}_P, x) = \|\tilde{x}_P - x\|_2$, and measure the Pearson correlation coefficient (PCC) between the latent STD and $\mathcal{R}_{L_2}(\{i\})$ to see if there is any similar ordering effect. If this coefficient is close to 1, then the latent STD can empirically serve as an indicator of the importance of each latent dimension, just like in PCA.

## 4 RELATED WORKS AND LIMITATIONS

The usual way of encouraging the latent set to be low dimensional is sparse coding (Bengio et al., 2013), which applies sparsity penalties like Lasso (Tibshirani, 1996; Ng, 2004; Lee et al., 2006), Student's t-distribution (Olshausen & Field, 1996; Ranzato et al., 2007), KL divergence (Le et al., 2011; Ng et al., 2011) *etc.*, over the latent code vector to induce as many zero-features as possible. However, as discussed in §2.3, a latent representation transformed by a homeomorphism $h$ is equivalent to the original one in the sense that $g_\theta \circ e_\theta(x) = (g_\theta \circ h^{-1}) \circ (h \circ e)(x)$, so translating the sparse-coding latent set arbitrarily—which makes the zero-features no longer zero—provides us an equally good representation. This equivalent latent set is then one that has zero-STDs along many latent dimensions, which is what this work tries to construct. Yet $h$ is not restricted to translation. For instance, rotation is also homeomorphic, so rotating that flat latent set also gives an equivalently good representation. IRMAE (Jing et al., 2020) can be regarded as a case of this, in which the latent

set is implicitly compressed into a linear subspace not necessarily aligned with the latent coordinate axes. There are also stochastic methods like K-sparse AE (Makhzani & Frey, 2013) that compress the latent set by randomly dropping out inactive latent dimensions during training.

People also care about the information content each latent dimension contains, and hope the latent dimensions can be ordered by their degrees of importance. Nested dropout (Rippel et al., 2014) is a probabilistic way that makes the information content in each latent dimension decrease as the latent dimension index increases. PCA-AE (Pham et al., 2022) achieves a similar effect by gradually expanding the latent space while reducing the covariance loss. Our work differs given that least volume is deterministic, requiring no multistage training, and we only require the information content to decrease with the STD of the latent dimension instead of the latent dimension's index number. Some researchers have also investigated the relationship between PCA and linear autoencoders (Rippel et al., 2014; Plaut, 2018; Kramer, 1991). In recent years more researchers started to look into preserving additional information on top of topological properties in the latent space (Moor et al., 2020; Trofimov et al., 2023; Gropp et al., 2020; Chen et al., 2020; Yonghyeon et al., 2021; Nazari et al., 2023), such as geometric information. A detailed methodological comparison between least volume and some aforementioned methods is given in Table 1 to illustrate its distinction.

Table 1: Comparison of Methods that Automatically Reduce Autoencoder Latent Dimensions

| Method | Least Volume | Nested Dropout | PCA-AE | IRMAE | K-sparse AE |
|---|---|---|---|---|---|
| Deterministic? | ✓ | ✗ | ✓ | ✓ | ✗ |
| Nonlinear AE? | ✓ | ✓ | ✓ | ✓ | ✗ |
| Penalty Term? | ✓ | ✗ | ✗ | ✗ | ✗ |
| Single-stage Training? | ✓ | ✓ | ✗ | ✓ | ✓ |
| Importance Ordering? | ✓ | ✓ | ✓ | ✗ | ✗ |

It is worth noting that this work is discussed under the scope of *continuous* autoencoders. Recently VQ-VAE (Van Den Oord et al., 2017) has attracted a lot of attention for its efficacy. Although it is modelled by continuous neural networks, its encoding operation is not continuous, because mapping the encoder's output to the nearest embedding vector is discontinuous. Hence this work cannot be readily applied to it. More discussions about LV's other technical limitations are included in §C.1.

## 5 EXPERIMENTS

In this section, we demonstrate the primary experiment results that reflect the properties and performance of Least Volume. We examine LV's dimension reduction and ordering effect on benchmark image datasets, and conduct ablation study on the volume penalty and the decoder's Lipschitz regularization. Due to the space limit, we instead present in §B of the appendix more details about these experiments, additional examinations, applications to downstream tasks and some toy problems for illustrating the effect of LV.

### 5.1 IMAGE DATASETS

In this experiment, we compare the performance of different latent regularizers that have the potential of producing a compressed latent subspace. Specifically, these four regularizers are: $L_1$ norm of the latent code (denoted by "lasso"), $L_1$ norm of the latent STD vector (denoted by "l1"), volume penalty $L_{\text{vol}}$ with $\eta = 1$ (denoted by "vol" or "vol_e1.0") and the one in (Ranzato et al., 2007) based on student's t-distribution (denoted by "st"). We activate the spectral normalization on the decoder and apply these four regularizers to a 5D synthetic image dataset (detailed in §B.2.2 of the appendix), the MNIST dataset (Deng, 2012) and the CIFAR-10 dataset (Krizhevsky et al., 2014), then investigate how good they are at reducing the latent set's dimensionality without sacrificing reconstruction performance, and whether there is any high correlation between the latent STD and the degree of importance (We also additionally perform these experiments on the CelebA dataset (Liu et al., 2015), which is included in the appendix §B.2.1 for space reasons). For these image datasets,

we use latent spaces of 50, 128, and 2048 dimensions respectively. For each case, we then conduct the experiment over a series of $\lambda$ ranging from 0.03 to 0.0001 with three cross validations and use the result's STD as the error bars. We normalize all pixels values between 0 and 1 and use binary cross entropy as the reconstruction loss $J$ for all experiments, not only because in practice it has faster convergence rate than the usual MSE loss, but also because minimizng it in the $[0, 1]$ range is equivalent to minimizing the other norm-based reconstruction loss. All experiments are performed on NVIDIA A100 SXM GPU 80GB. More tips regarding hyperparameter tuning, model training and usage can be found in §C of the appendix.

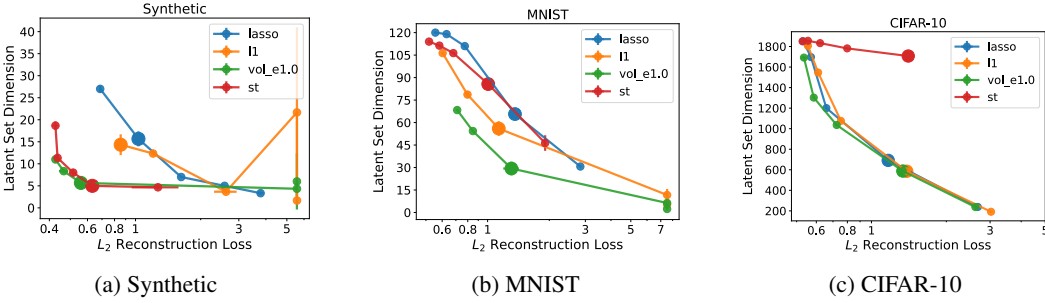

(a) Synthetic  (b) MNIST  (c) CIFAR-10

Figure 2: Latent Set Dimensionality vs $L_2$ Reconstruction Error. The markers corresponds to the records obtained at different $\lambda$ listed in Table B.5. In general, the higher the $\lambda$, the lower the latent set dimension, but the reconstruction error also increases accordingly.

Comparing these regularizers' dimension reduction performance is not a straightforward task. This is because each latent regularizer $R$, when paired with the reconstruction loss $J$ via $L = J + \lambda \cdot R$, creates its own loss landscape, such that the same $\lambda$ leads to different $J$ for different $R$ after optimizing $L$. This means under the same $\lambda$, some regularizers may choose to compress the latent set more simply by sacrificing the reconstruction quality more, so we cannot compare these regularizers under the same $\lambda$. However, because adjusting $\lambda$ is essentially trading off between reconstruction and dimension reduction, we can plot *the dimension reduction metric against the reconstruction metric* for comparison. An efficient $R$ should compress the latent set more for the same reconstruction quality. Figure 2 plots the latent set dimension against the autoencoder's $L_2$ reconstruction error, where the latent set dimension is the number of dimensions left after cumulatively pruning the latent dimensions with the smallest STDs until their joint explained reconstruction exceeds $1\%$. We can see that for all datasets, the volume penalty $L_{\mathrm{vol}}$ always achieves the highest compression when the $L_2$ reconstruction error is reasonable (empirically that means $< 2$. See §B.3.1 in the appendix).

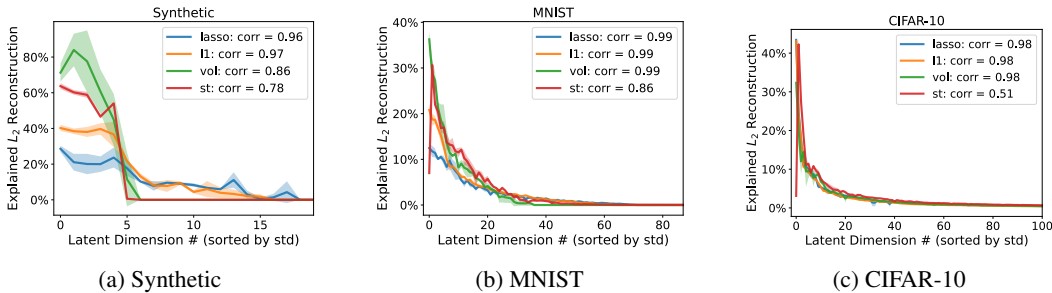

(a) Synthetic  (b) MNIST  (c) CIFAR-10

Figure 3: Explained Reconstruction vs Latent Dimension # (sorted in descending order of STD)

For each dataset, we select three autoencoding cases with comparable reconstruction quality respectively for the four regularizers (marked by large dots in Fig. 2). Then in Fig. 3, for each case we sort the latent dimensions in descending order of STD value, and plot their individual explained reconstructions against their ordered indices. For all regularizers, we see that the explained $L_2$ reconstruction generally increases with the latent STD, although it is not perfectly monotonic. Nevertheless, we can conclude that the explained reconstruction is highly correlated with latent STD, since the PCCs are all close to 1 (except for "st" based on student's t-distribution, which has a sudden drop as the dimension index reach 0), as shown in Fig. 3.

## 5.2 ABLATION STUDY

We first investigate the necessity of the Lipschtiz regularization on the decoder. For each experiment we keep everything the same except switching off the spectral normalization. Figure 4 shows that without Lipschitz regularization, the latent set's dimensionality cannot be effectively reduced. This is indeed due to the isotropic shrinkage as mentioned in §2.2, as the latent dimensions of the autoencoders without Lipschitz regularization have orders of magnitude lower STD values (see §B.3.4). In addition, Fig. 5 shows that apart from the Lipschitz regularization, the regularization on latent set is also necessary for reducing the latent set's dimension. The Lipschitz regularization alone does not induce any dimension reduction effect.

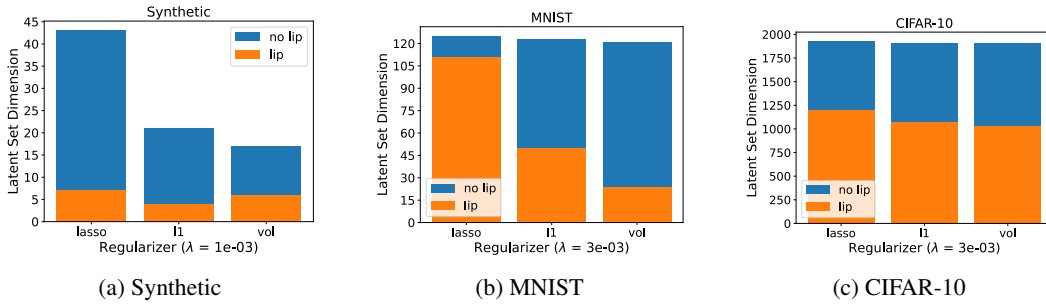

(a) Synthetic        (b) MNIST        (c) CIFAR-10

Figure 4: Latent Set Dimensionality with and without Lipschitz Constraint on Decoder.

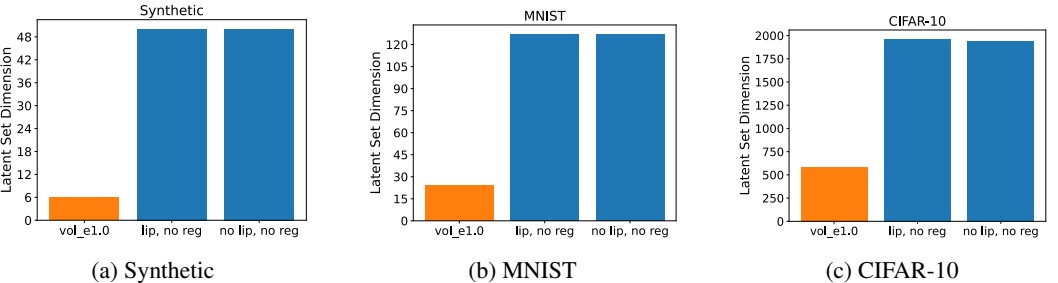

(a) Synthetic        (b) MNIST        (c) CIFAR-10

Figure 5: Latent Set Dimensionality with and without Volume Penalty

## 6 CONCLUSION

This work introduces Least Volume regularization for continuous autoencoders. It can compress the latent representation of a dataset into a low dimensional latent subspace for easy extraction. We show that minimizing the volume is equivalent to minimizing an upper bound of the latent code's generalized variance, and that using a $K$-Lipschitz decoder prevents the latent set's isotropic shrinkage. This approach achieves better dimension reduction results than several $L_1$ distance-based counterparts on multiple benchmark image datasets. We further prove that PCA is a linear special case of the least volume formulation, and when least volume is applied to nonlinear autoencoders, it demonstrates a similar ordering effect, such that the STD of each latent dimension is highly correlated with the latent dimension's importance.

### ACKNOWLEDGMENTS

We acknowledge the support from the National Science Foundation through award #1943699 as well as ARPA-E award DE-AR0001216.

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

## A  PROOFS AND REMARKS FOR §3

### A.1  PROOFS FOR §3.1

**Theorem 1.** *The square of volume—i.e., $\prod_i \sigma_i^2$ —is a tight upper bound of the latent determinant* $\det S$. *If $\det S$ is lower bounded by a positive value, then $\prod_i \sigma_i^2$ is minimized down to $\det S$ if and only if $S$ is diagonal.*

*Proof.* The diagonal entries of positive semi-definite matrices are always non-negative, and Cholesky decomposition states that any real positive definite matrix $S$ can be decomposed into $S = LL^\top$ where $L$ is a real lower triangular matrix with positive diagonal entries. So $\det S = (\det L)^2 = \prod_i [L]_{ii}^2 \le \prod_i \|\mathbf{l}_i\|_2^2 = \prod_i [S]_{ii} = \prod_i \sigma_i^2$, where $\mathbf{l}_i$ is the $i$th row of $L$.

Hence for a positive definite $S$, the inequality can only be reduced to equality when $L$ is diagonal, which in turn makes $S$ diagonal. □

### A.1.1  EXAMPLE: PITFALL OF $L_1$

We can use a simple example to show that the volume is better than $L_1$-based regularizers in terms of determining the sparsity of $\boldsymbol{\sigma}$, *i.e.*, how flat the latent set is. Figure A.1 shows a straight orange line $\mathcal{Z}_l$ of length $\frac{3}{2}\pi$ and a blue arc $\mathcal{Z}_a$ of the same length. Clearly there exist an isometry $h : \mathcal{Z}_l \to \mathcal{Z}_a$. Suppose the arc is a latent set created through $\mathcal{Z}_a = e(\mathcal{X})$, then $\mathcal{Z}_l = h^{-1} \circ e(\mathcal{X})$ is an equivalently good one, provided that the encoder $e$ has enough complexity to learn $h^{-1} \circ e$. Moreover, because the isometry $h$ is 1-Lipschitz, then a $K$-Lipschitz decoder $g$ with enough complexity can also learn $g \circ h$ since this is also $K$-Lipschitz, so both $\mathcal{Z}_l$ and $\mathcal{Z}_a$ are latent sets that can be produced by $e$ and $g$ with enough complexity. Now we introduce uniform distribution over these two sets and evaluate $\|\boldsymbol{\sigma}\|_1$ and $\prod_i \sigma_i$ on them. Figure A.1 shows that only the volume $\prod_i \sigma_i$ correctly tells the line $\mathcal{Z}_l$ is flatter.

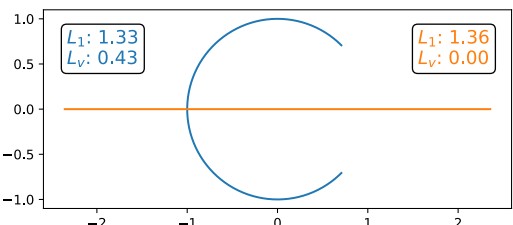

Figure A.1: A pedagogical example where minimizing the $L_1$ regularizers produces less flattened representations than minimizing the volume. Here "$L_1$" refers to $\|\boldsymbol{\sigma}\|_1$ and "$L_v$" refers to $\prod_i \sigma_i$.

### A.2  PROOFS FOR §3.2

**Lemma 1.** *If $\forall x \in \mathcal{X}$, $g \circ e(x) = x$, then both $e$ and $g$ are bijective between $\mathcal{X}$ and $\mathcal{Z} = e(\mathcal{X})$.*

*Proof.* $e|_{\mathcal{X}}$ is injective because for any pair $\{x, x'\} \subseteq \mathcal{X}$ that satisfies $e(x) = e(x')$, we have $x = g \circ e(x) = g \circ e(x') = x'$. It is then bijective because $e|_{\mathcal{X}}$ is surjective onto $\mathcal{Z}$ by definition. $g|_{\mathcal{Z}}$ is surjective onto $\mathcal{X}$ because $g(\mathcal{Z}) = g \circ e(\mathcal{X}) = \mathcal{X}$. It is then injective thus bijective because $\forall z = e(x) \in \mathcal{Z}, e \circ g(z) = e \circ g \circ e(x) = e(x) = z$. □

**Theorem 2.** *A continuous autoencoder between the data space $X = \mathbb{R}^n$ and the latent space $Z = \mathbb{R}^m$ with the norm-based reconstruction error $\|g \circ e(x) - x\| = 0$ everywhere on the dataset $\mathcal{X} \subseteq X$ learns a topological embedding of the dataset. In other words, the latent set $\mathcal{Z} = e(\mathcal{X}) \subseteq Z$ is a homeomorphic copy of the dataset.*

*Proof.* Due to the positive definiteness of norm and Lemma 1, both the encoder restriction $e|_{\mathcal{X}}$ and the decoder restriction $g|_{\mathcal{Z}}$ are bijective functions between $\mathcal{X}$ and $\mathcal{Z}$. Because both the encoder $e : X \to Z$ and the decoder $g : Z \to X$ are continuous, their restrictions are also continuous (in terms of the subspace topology). Since $g|_{\mathcal{Z}}$ is then the continuous inverse function of $e|_{\mathcal{X}}$, by definition $e|_{\mathcal{X}}$ is a topological embedding (Lee, 2010). □

**Corollary 2.1.** *If the above errorless autoencoder's latent set has the form $\mathcal{Z} = \mathcal{Z}' \times c$ where $c \in \mathbb{R}^p$ is constant and $\mathcal{Z}' \subseteq Z' = \mathbb{R}^{m-p}$, then $\pi \circ e|_{\mathcal{X}}$ is also a topological embedding, where $\pi : Z' \times c \to Z'$ is the projection map.*

*Proof.* $\pi$ is homeomorphic because it is bijective, while continuous and open in terms of the product topology. Thus $\pi \circ e|_{\mathcal{X}}$ is also homeomorphic onto its image $\mathcal{Z}'$. $\qquad\square$

**Remark.** *Because $\mathcal{Z}'$ is homeomorphic to $\mathcal{X}$, the subspace dimension $\dim Z' = m - p$ cannot be lower than $\dim \mathcal{X}$, as otherwise it violates the topological invariance of dimension (Lee, 2012). So we need not to worry about extravagant scenarios like "$\mathcal{Z}'$ collapsing into a line while $\mathcal{X}$ is a surface". This is not guaranteed for a discontinuous autoencoder since it does not learn a topological embedding.*

## A.3 PROOFS FOR §3.3

**Theorem 3.** *Suppose the STD of latent dimension $i$ is $\sigma_i$. If we fix $z_i$ at its mean $\bar{z}_i$ for each $i \in P$ where $P$ is the set of indices of the dimensions we decide to prune, then the $L_2$ reconstruction error of the autoencoder with a $K$-Lipschitz decoder $g$ increases by at most $K\sqrt{\sum_{i \in P} \sigma_i^2}$.*

*Proof.* Suppose the reconstruction error $\epsilon$ is measured by $L_2$ distance in the data space as $\epsilon = \mathbb{E}\|x - \hat{x}\| = \mathbb{E}\|x - g \circ e(x)\| = \mathbb{E}\|x - g(z)\|$. Let the new reconstruction error after latent pruning be $\tilde{\epsilon} = \mathbb{E}\|x - \tilde{x}_P\| = \mathbb{E}\|x - g \circ p_P \circ e(x)\| = \mathbb{E}\|x - g(\tilde{z}_P)\|$, where $p_P$ is the pruning operator defined by $[\tilde{z}_P]_i = [p_P(z)]_i = \begin{cases} z_i & i \notin P \\ \bar{z}_i & i \in P \end{cases}$. Then due to the subadditivity of norm and the $K$-Lipschitz continuity of $g$ we have

$$\tilde{\epsilon} - \epsilon \leq \mathbb{E}\|\hat{x} - \tilde{x}_P\| = \mathbb{E}\|g(\tilde{z}_P) - g(z)\|$$

$$\leq K \cdot \mathbb{E}\|\tilde{z}_P - z\| = K\sqrt{\mathbb{E}\left[\|\tilde{z}_P - z\|^2\right] - \mathrm{Var}\left[\|\tilde{z}_P - z\|\right]} \tag{7}$$

$$\leq K\sqrt{\mathbb{E}\left[\|\tilde{z}_P - z\|^2\right]} = K\sqrt{\sum_{i \in P}\mathbb{E}\left[(z_i - \bar{z}_i)^2\right]} = K\sqrt{\sum_{i \in P}\sigma_i^2} \tag{8}$$

Of course, (7) is an upper bound tighter than (8), but the minimalistic (8) also conveys our idea clearly. $\qquad\square$

## A.4 PROOFS FOR §3.4

**Lemma 2.** *Suppose for a symmetric matrix $S$ we have $A^\top S A = \Sigma$ and $A^\top A = I$, where $\Sigma$ is a diagonal matrix whose diagonal elements are the largest eigenvalues of $S$. Then $A$'s columns consist of the orthonormal eigenvectors corresponding to the largest eigenvalues of $S$.*

*Proof.* Denote the eigenvalues of $S$ by $\sigma_1 \geq \sigma_2 \geq \cdots$, the corresponding orthonormal eigenvectors by $u_1, u_2, \cdots$, and the $i$th column vector of $A$ by $a_i$. Without any loss of generality, we also assume that the diagonal elements of $\Sigma$ are sorted in descending order so that $\Sigma_{ii} = \sigma_i$, as the following proof is essentially independent of the specific order.

We can prove this lemma by induction. To begin with, suppose for $a_1$ we have $a_1^\top S a_1 = \sigma_1$, then $a_1$ must fall inside the eigenspace $E_{\sigma_1}$ spanned by the eigenvector(s) of $\sigma_1$. Assume this is not true, then $a_1 = v + v_\perp$ where $v \in E_{\sigma_1}$, $v_\perp \perp E_{\sigma_1}$, $\|v_\perp\| \neq 0$ and $\|a_1\|^2 = \|v\|^2 + \|v_\perp\|^2 = 1$. It follows that $a_1^\top S a_1 = v^\top S v + v_\perp^\top S v_\perp < \sigma_1\|v\|^2 + \sigma_1\|v_\perp\|^2 = \sigma_1$, which violates the assumption. The inequality holds because we have the decomposition $v_\perp = \sum_{i \in \mathcal{I}} \alpha_i u_i$, $\mathcal{I} = \{i \mid u_i \perp E_{\sigma_1}\}$ where $u_i$'s corresponding $\sigma_i$ is smaller than $\sigma_1$, so $v_\perp^\top S v_\perp = \sum_{i \in \mathcal{I}} \sigma_i \alpha_i^2 < \sigma_1 \sum_{i \in \mathcal{I}} \alpha_i^2 = \sigma_1\|v_\perp\|^2$.

Now assume that for $a_k$, its predecessors $a_1 \ldots a_{k-1}$ respectively all fall inside the corresponding eigenspaces $E_{\sigma_1} \ldots E_{\sigma_{k-1}}$. Since $a_k$ is orthogonal to $a_1 \ldots a_{k-1}$, we can only find $a_k$ in $\sum_{i \geq k} E_{\sigma_i}$. Then following the same rationale as in the $a_1$ case, $a_k$ must be inside $E_{\sigma_k}$. $\qquad\square$

**Proposition 1.** *An autoencoder recovers the principal components of a dataset $X$ after minimizing the volume, if we:*

1. *Make the encoder $e(x) = Ax + a$ and the decoder $g(z) = Bz + b$ linear maps,*

2. *Force the reconstruction error $\|g \circ e(X) - X\|_F$ to be strictly minimized,*

3. *Set the Lipschitz constant to 1 for the decoder's regularization,*

4. *Assume the rank of the data covariance $\text{Cov}(X)$ is not less than the latent space dimension $m$.*

*Specifically, $B$'s columns consist of the principal components of $X$, while $A$ acts exactly the same as $B^\top$ on $X$. When $X$ has full row rank, $A = B^\top$.*

*Proof.* According to (Bourlard & Kamp, 1988), minimizing the reconstruction error cancels out the biases $a$ and $b$ together, and simplifies the reconstruction loss into $\|BAX' - X'\|_F$ where $X' = X - \bar{X}$. Therefore, for simplicity and without any loss of generality, hereafter we assume $\bar{X} = 0$, $e(x) = Ax$ and $g(z) = Bz$. Given a dataset $X \in \mathbb{R}^{d \times n}$ of $n$ samples, condition #1 means the corresponding latent codes are $Z = AX$, whose covariance matrix is $S = \frac{1}{n-1} ZZ^\top = \frac{1}{n-1} AXX^\top A^\top = AS_X A^\top$ where $S_X$ is the data covariance.

Condition #2 necessitates that both the encoder $A$ and the decoder $B$ possess full-rank. This comes from the *Eckart–Young–Mirsky theorem* (Eckart & Young, 1936) and the inequality $\text{rank}(AB) \leq \min(\text{rank}(A), \text{rank}(B))$. Moreover, for a full-rank decoder $B = V\Sigma^{-1}U^\top$ (here for simplicity we use the SVD formulation where $\Sigma$ is diagonal), the encoder $A$ must satisfy $AX = B^\dagger X = U\Sigma V^\top X$ when we minimize the reconstruction loss $\|BAX - X\|_F$, so the reconstruction loss reduces into $\|VV^\top X - X\|_F$. From the minimum-error formulation of PCA (Bishop & Nasrabadi, 2006), we know $V$ should be of the form $V = PQ$, where $Q^\top Q = QQ^\top = I$ and $P$'s column vectors are the orthonormal eigenvectors corresponding to the $m$ largest eigenvalues of the data covariance $S_X$, such that $P^\top S_X P = \Lambda$ where $\Lambda$ is a diagonal matrix of the $m$ largest eigenvalues. Therefore we can only minimize the volume by changing $U$, $\Sigma$ and $Q$.

It follows that $S = AS_X A^\top = U\Sigma V^\top S_X V\Sigma^\top U^\top = U\Sigma Q^\top \Lambda Q\Sigma^\top U^\top$. Because according to Theorem 1, minimizing the volume $\prod_i^m \sqrt{[S]_{ii}}$ minimizes the determinant $\det S = (\det \Sigma)^2 \det \Lambda$, then under condition #3 and #4, this is minimized only when $[\Sigma]_{ii} = 1$ for all $i$, given that the decoder is required to be 1-Lipschitz so $[\Sigma]_{ii} \geq 1$ for all $i$, and $\det \Lambda > 0$. Hence $\Sigma = I$, $B^\dagger = B^\top$ and $B$ has orthonormal columns because $B^\top B = U\Sigma^{-2}U^\top = I$.

Therefore $S^\star$—the volume-minimized $S$—satisfies $S^\star = UQ^\top \Lambda QU^\top$ and thus is orthogonally similar to $\Lambda$. Since $S^\star$ is diagonal (Theorem 1), it must have the same set of diagonal entries as those of $\Lambda$, though not necessarily in the same order. Thus when the volume is minimized, we have $AS_X A^\top = B^\dagger S_X (B^\dagger)^\top = B^\top S_X B = S^\star$. Because $B$'s columns are orthonormal, according to Lemma 2, this identity only holds when $B$'s column vectors are the eigenvectors corresponding to the $m$ largest eigenvalues of $S_X$, so the linear decoder $B$ consists of the principal components of $X$. $\square$

**Remark 1.1.** *Although $A$ is not necessarily equal to $B^\top$, it projects $X$ into the latent space the same way as $B^\top$. It can also be verified that it must have the form $A = B^\top + W$ where each row of $W$ lies in $\ker X^\top$. This means the identity $A = B^\top$ holds when $X$ has full row rank, in which case $A$ also recovers the principal components of $X$.*

**Remark 1.2.** *It is easy to check that setting the decoder's Lipschitz constant to any value $K$ will just scale down all the eigenvalues by $K^2$, without hurting any of the principal dimension aligning of PCA or distorting the ratio between dimensions.*

**Remark 1.3.** *This proof indicates that minimizing the volume is disentangled from reducing the reconstruction loss, at least for the linear case. Specifically, the reconstruction loss controls only $P$, while the least volume regularization controls the rotation and scaling in the latent space respectively through $U$, $Q$ and $\Sigma$. The product $U\Sigma Q^\top$ then models the family $\mathcal{H}$ of linear homeomorphisms that the linear autoencoder (without bias vectors) can additionally learn beyond preserving information through $P^\top$, as discussed in §2.3. Indeed, the linear encoder is of the form $A = (U\Sigma Q^\top) \circ P^\top$ while the linear decoder is of the form $B = P \circ Q\Sigma^{-1}U^\top = P \circ (U\Sigma Q^\top)^{-1}$.*

*Due to the minimization of volume, $\Sigma$ ends up being an identity matrix, making both $g$ and $e$ isometric over the dataset $X$. This means $e$ does not scale up or down any data dimension in the latent space. We argue that this is the primary reason why each PCA latent dimension's degree of importance*

*scales with its variance, because one can easily check that as long as $\Sigma = I$, Proposition 2 below still holds for an "imperfect" PCA where $A = B^\top = UQ^\top P^\top$. We may expect a similar ordering effect for nonlinear autoencoders.*

*The rotations $U$ and $Q$, on the other hand, help cram as much data information into the most primary latent dimensions as possible, such that when we remove any number of the least informant latent dimensions, the remaining dimensions still store the most amount of information they can extract from $X$ in terms of minimizing the reconstruction loss. For nonlinear autoencoders, we may expect a cramming effect that is similar yet not identical, because after stripping off some least informant latent dimensions, the nonlinear autoencoder might further reduce the reconstruction loss by curling up the decoder's image—now of lower dimension—in the dataset to traverse more data region.*

**Proposition 2.** *Let $\lambda_i$ be the eigenvalues of the data covariance matrix. The explained variance $\frac{\lambda_i}{\sum_j^n \lambda_j}$ of a given latent dimension $i$ of a linear PCA model is the ratio between $\mathcal{E}_{\|\cdot\|_2^2}(\{i\})$—the MSE reconstruction error induced by pruning this dimension (i.e., setting its value to its mean $0$) and $\mathcal{E}_{\|\cdot\|_2^2}(\Omega)$—the one induced by pruning all latent dimensions, where $\Omega$ denotes the set of all latent dimension indices.*

*Proof.* First we show that the $i$th eigenvalue $\lambda_i$ of the data covariance matrix equals the MSE introduced by pruning principal dimension $i$. Let $v_i$ denote the eigenvector corresponding to the $i$th principal dimension. Then pruning this dimension leads to an incomplete reconstruction $\tilde{x}_{\{i\}}$ that satisfies:

$$\mathcal{E}_{\|\cdot\|_2^2}(\{i\}) := \mathbb{E}[\|\tilde{x}_{\{i\}} - x\|_2^2] = \mathbb{E}[\|\langle x, v_i\rangle \cdot v_i\|_2^2] = \mathbb{E}[\langle x, v_i\rangle^2] = \lambda_i \tag{9}$$

Next we show that the $\mathcal{E}$ of linear PCA has additivity, *i.e.*, the joint MSE induced by pruning several dimensions in $P$ altogether equals the sum of their individual induced MSE:

$$\begin{aligned}
\mathcal{E}_{\|\cdot\|_2^2}(P) := \mathbb{E}[\|\tilde{x}_P - x\|_2^2] &= \mathbb{E}[\|\textstyle\sum_{i\in P}\langle x, v_i\rangle \cdot v_i\|_2^2] \\
&= \mathbb{E}[\langle \textstyle\sum_{i\in P}\langle x, v_i\rangle \cdot v_i, \ \sum_{i\in P}\langle x, v_i\rangle \cdot v_i\rangle] \\
&= \mathbb{E}[\textstyle\sum_{i\in P}\langle x, v_i\rangle^2] = \sum_{i\in P}\mathbb{E}[\langle x, v_i\rangle^2] \\
&= \textstyle\sum_{i\in P}\lambda_i = \sum_{i\in P}\mathcal{E}_{\|\cdot\|_2^2}(\{i\})
\end{aligned} \tag{10}$$

Hence the claim is proved. $\qquad\square$

**Remark 2.1.** *The subtraction in Eqn. 6 is a generalization of the case where the PCA model's number of components is set smaller than the data covariance's rank $r$. In this case, the PCA model—as a linear AE—will inevitably have a reconstruction error $\epsilon$ even if no latent dimension is pruned. Yet we may regard this inherent error $\epsilon = \mathbb{E}[\|\tilde{x}_I - x\|_2^2]$ as the result of implicitly pruning a fixed collection $I$ of additional $r - n$ latent dimensions required for perfect reconstruction. Then $\lambda_i$ in (9) can instead be explicitly expressed as $\lambda_i = \mathbb{E}[\|\tilde{x}_{\{i\}\cup I} - x\|_2^2] - \epsilon$ to accommodate this implicit pruning. Therefore the induced reconstruction error $\mathcal{E}_{\|\cdot\|_2^2}(\{i\})$ is actually the change in reconstruction error before and after pruning $i$ (with the inherent reconstruction error $\epsilon$ taken into account). The one in Eqn. 6 is thus a generalization of this, using the change in reconstruction error to indicate each dimension's importance.*

# B  TECHNICAL DETAILS OF EXPERIMENTS

## B.1  TOY PROBLEMS

We apply least volume to low-dimensional toy datasets to pedagogically illustrate its effect. In each case, we make the latent space dimension equal to the data space dimension. Figure B.2 shows that the autoencoders regularized by least volume successfully recover the low dimensionality of the 1D and 2D data manifold respectively in the latent spaces by compressing the latent sets into low dimensional latent subspaces, without sacrificing the reconstruction quality. Not only that, with a large enough weight $\lambda$ for the volume penalty, in the noisy 2D problem, the autoencoder also manages to remove the data noise perpendicular to the 2D manifold (Fig. B.2b.ii).

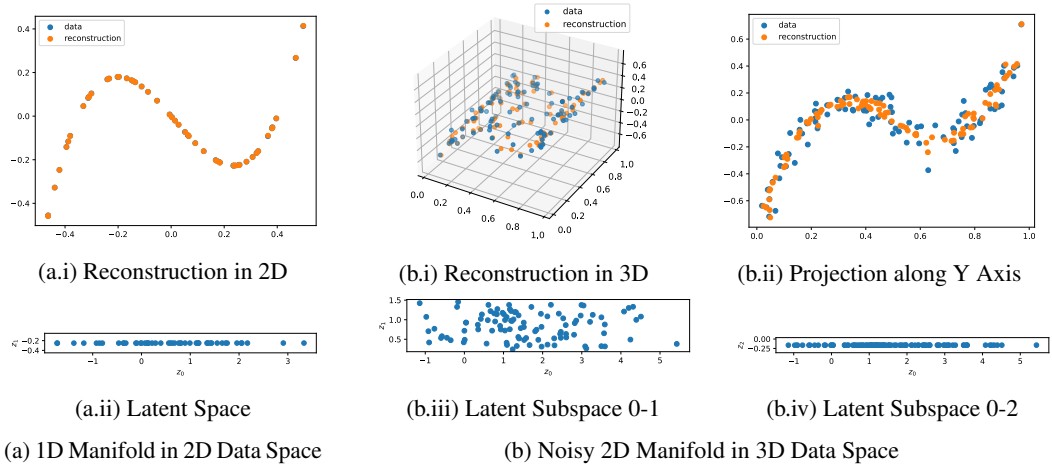

(a.i) Reconstruction in 2D   (b.i) Reconstruction in 3D   (b.ii) Projection along Y Axis

(a.ii) Latent Space   (b.iii) Latent Subspace 0-1   (b.iv) Latent Subspace 0-2

(a) 1D Manifold in 2D Data Space   (b) Noisy 2D Manifold in 3D Data Space

Figure B.2: Least volume on low dimensional toy problems.

### B.1.1 DATASET

The 1D manifold dataset $\{(x, y)\}$ consists of 50 samples, where $x$ are uniformly sampled from $[-0.5, 0.5]$ and $y$ are created via $y = 10x(x - 0.4)(x + 0.35)$.

The 2D manifold dataset $\{(x, y, z)\}$ consists of 100 samples, where $x$ and $y$ are uniformly sampled from $[-0.5, 0.5] \times [-0.5, 0.5]$, and $z$ are created via $z = 10x(x - 0.3)(x + 0.3)$. After this, we add Gaussian noise $\mathcal{N}(\mu = 0, \sigma = 0.1)$ to the $z$ dimension

### B.1.2 ARCHITECTURE

In Table B.1 we list the architectures of the autoencoders for both toy datasets. Here $\text{Linear}_n$ denotes a fully connected layer of $n$ output features, while $\text{SN-Linear}_n$ denotes one regularized by spectral normalization. $\text{LeakyReLU}_\alpha$ denotes a LeakyReLU activation function with negative slope $\alpha$.

Table B.1: Architectures of Autoencoders

| Manifold | Encoder | Decoder |
|---|---|---|
| 1D | $x \in \mathbb{R}^2$
$\to \text{Linear}_{32} \to \text{LeakyReLU}_{0.2}$
$\to \text{Linear}_{32} \to \text{LeakyReLU}_{0.2}$
$\to \text{Linear}_{32} \to \text{LeakyReLU}_{0.2}$
$\to \text{Linear}_{32} \to \text{LeakyReLU}_{0.2}$
$\to \text{Linear}_2 \to z \in \mathbb{R}^2$ | $z \in \mathbb{R}^2$
$\to \text{SN-Linear}_{32} \to \text{LeakyReLU}_{0.2}$
$\to \text{SN-Linear}_{32} \to \text{LeakyReLU}_{0.2}$
$\to \text{SN-Linear}_{32} \to \text{LeakyReLU}_{0.2}$
$\to \text{SN-Linear}_{32} \to \text{LeakyReLU}_{0.2}$
$\to \text{SN-Linear}_2 \to x \in \mathbb{R}^2$ |
| 2D | $x \in \mathbb{R}^3$
$\to \text{Linear}_{128} \to \text{LeakyReLU}_{0.2}$
$\to \text{Linear}_{128} \to \text{LeakyReLU}_{0.2}$
$\to \text{Linear}_{128} \to \text{LeakyReLU}_{0.2}$
$\to \text{Linear}_{128} \to \text{LeakyReLU}_{0.2}$
$\to \text{Linear}_3 \to z \in \mathbb{R}^3$ | $z \in \mathbb{R}^3$
$\to \text{SN-Linear}_{128} \to \text{LeakyReLU}_{0.2}$
$\to \text{SN-Linear}_{128} \to \text{LeakyReLU}_{0.2}$
$\to \text{SN-Linear}_{128} \to \text{LeakyReLU}_{0.2}$
$\to \text{SN-Linear}_{128} \to \text{LeakyReLU}_{0.2}$
$\to \text{SN-Linear}_3 \to x \in \mathbb{R}^3$ |

### B.1.3 HYPERPARAMETERS

The hyperparameters are listed in Table B.2. The reconstruction loss is MSE, the volume penalty is $\sqrt[m]{\prod_i \sigma_i}$, and the optimizer is Adam.

Table B.2: Hyperparameters

| Dataset | Batch Size | $\lambda$ | $\eta$ | Learning Rate | Epochs |
|---------|-----------|-----------|--------|---------------|--------|
| 1D | 50 | 0.001 | 0 | 0.001 | 10000 |
| 2D | 100 | 0.01 | 0 | 0.0001 | 10000 |

## B.2 IMAGE DATASETS

### B.2.1 CELEBA RESULT

We additionally perform the dimension reduction experiment in §5.1 on the CelebA dataset.

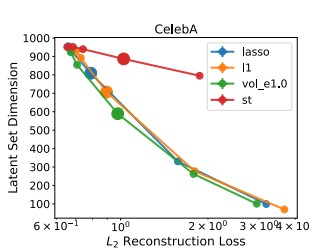 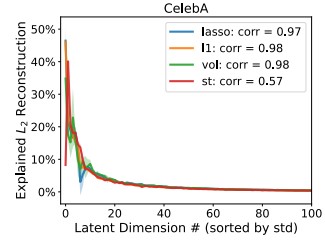 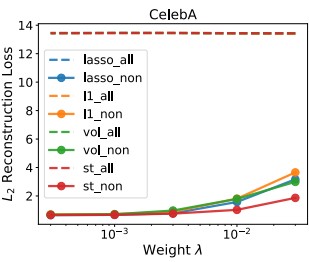

(a) Latent Set Dimensionality vs $L_2$ Reconstruction Error

(b) Explained Reconstruction vs Latent Dimension #

(c) $L_2$ Reconstruction vs $\lambda$ (more in §B.3.1)

Figure B.3: Dimension Reduction Result of Sparse Regularizers on CelebA

### B.2.2 DATASETS

The 5D synthetic dataset is produced by feeding an image of a red circle to Torchvision's RandomAffine and ColorJitter layers to randomly generate images of transformed circles of different colors, then resizing them to $32 \times 32 \times 3$. Specifically, we set their parameters to translate=(0.2, 0.2), scale=(0.2, 0.5), brightness=(0.3, 1), hue=(0, 0.5), corresponding to 3 spatial dimensions ($x$-translation, $y$-translation, circle size) and 2 color dimensions (hue and brightness). The other information of the three image sets are listed in Table B.3. All pixel values are normalized to between 0 and 1.

Table B.3: Dataset Information

| Dataset | Synthetic | MNIST | CIFAR-10 | CelebA |
|---------|-----------|-------|----------|--------|
| Training Set Size | 30000 | 60000 | 50000 | 162770 |
| Test Set Size | 6000 | 10000 | 10000 | 39829 |
| Image Dimension | $32 \times 32 \times 3$ | $32 \times 32 \times 1$ | $32 \times 32 \times 3$ | $32 \times 32 \times 3$ |

### B.2.3 ARCHITECTURE

In Table B.4 we list the architectures of the autoencoders for these datasets. Here $\text{Conv}_n$ denotes convolutional layer of $n$ output channels, while SN-Deconv$_n$ denotes spectral-normalized deconvolutional layer of $n$ output channels. For all such layers we set kernel size to 4, stride to 2 and padding to 1.

### B.2.4 HYPERPARAMETERS

The hyperparameters are listed in Table B.5. The reconstruction loss is binary cross entropy and the optimizer is Adam. Only the volume penalty needs $\eta$. By default $\eta = 1$ for all experiments.

Table B.4: Architectures of Autoencoders

| Dataset | Encoder | Decoder |
|---------|---------|---------|
| Synthetic | $x \in \mathbb{R}^{32 \times 32 \times 3}$ 
 $\to \text{Conv}_{32} \to \text{LeakyReLU}_{0.2}$ 
 $\to \text{Conv}_{64} \to \text{LeakyReLU}_{0.2}$ 
 $\to \text{Conv}_{128} \to \text{LeakyReLU}_{0.2}$ 
 $\to \text{Conv}_{256} \to \text{LeakyReLU}_{0.2}$ 
 $\to \text{Reshape}_{2 \times 2 \times 256 \to 1024}$ 
 $\to \text{Linear}_{50} \to z \in \mathbb{R}^{50}$ | $z \in \mathbb{R}^{50} \to \text{SN-Linear}_{1024}$ 
 $\to \text{Reshape}_{1024 \to 2 \times 2 \times 256}$ 
 $\to \text{SN-Deconv}_{128} \to \text{LeakyReLU}_{0.2}$ 
 $\to \text{SN-Deconv}_{64} \to \text{LeakyReLU}_{0.2}$ 
 $\to \text{SN-Deconv}_{32} \to \text{LeakyReLU}_{0.2}$ 
 $\to \text{SN-Deconv}_{3} \to \text{Sigmoid}$ 
 $\to x \in \mathbb{R}^{32 \times 32 \times 3}$ |
| MNIST | $x \in \mathbb{R}^{32 \times 32 \times 1}$ 
 $\to \text{Conv}_{32} \to \text{LeakyReLU}_{0.2}$ 
 $\to \text{Conv}_{64} \to \text{LeakyReLU}_{0.2}$ 
 $\to \text{Conv}_{128} \to \text{LeakyReLU}_{0.2}$ 
 $\to \text{Conv}_{256} \to \text{LeakyReLU}_{0.2}$ 
 $\to \text{Reshape}_{2 \times 2 \times 256 \to 1024}$ 
 $\to \text{Linear}_{128} \to z \in \mathbb{R}^{128}$ | $z \in \mathbb{R}^{128} \to \text{SN-Linear}_{2048}$ 
 $\to \text{Reshape}_{2048 \to 4 \times 4 \times 128}$ 
 $\to \text{SN-Deconv}_{64} \to \text{LeakyReLU}_{0.2}$ 
 $\to \text{SN-Deconv}_{32} \to \text{LeakyReLU}_{0.2}$ 
 $\to \text{SN-Deconv}_{1} \to \text{Sigmoid}$ 
 $\to x \in \mathbb{R}^{32 \times 32 \times 1}$ |
| CIFAR-10 | $x \in \mathbb{R}^{32 \times 32 \times 3}$ 
 $\to \text{Conv}_{128} \to \text{LeakyReLU}_{0.2}$ 
 $\to \text{Conv}_{256} \to \text{LeakyReLU}_{0.2}$ 
 $\to \text{Conv}_{512} \to \text{LeakyReLU}_{0.2}$ 
 $\to \text{Conv}_{1024} \to \text{LeakyReLU}_{0.2}$ 
 $\to \text{Reshape}_{2 \times 2 \times 1024 \to 4096}$ 
 $\to \text{Linear}_{2048} \to z \in \mathbb{R}^{2048}$ | $z \in \mathbb{R}^{2048} \to \text{SN-Linear}_{8192}$ 
 $\to \text{Reshape}_{8192 \to 4 \times 4 \times 512}$ 
 $\to \text{SN-Deconv}_{256} \to \text{LeakyReLU}_{0.2}$ 
 $\to \text{SN-Deconv}_{128} \to \text{LeakyReLU}_{0.2}$ 
 $\to \text{SN-Deconv}_{3} \to \text{Sigmoid}$ 
 $\to x \in \mathbb{R}^{32 \times 32 \times 3}$ |
| CelebA | $x \in \mathbb{R}^{32 \times 32 \times 3}$ 
 $\to \text{Conv}_{128} \to \text{LeakyReLU}_{0.2}$ 
 $\to \text{Conv}_{256} \to \text{LeakyReLU}_{0.2}$ 
 $\to \text{Conv}_{512} \to \text{LeakyReLU}_{0.2}$ 
 $\to \text{Conv}_{1024} \to \text{LeakyReLU}_{0.2}$ 
 $\to \text{Reshape}_{2 \times 2 \times 1024 \to 4096}$ 
 $\to \text{Linear}_{1024} \to z \in \mathbb{R}^{1024}$ | $z \in \mathbb{R}^{1024} \to \text{SN-Linear}_{8192}$ 
 $\to \text{Reshape}_{8192 \to 4 \times 4 \times 512}$ 
 $\to \text{SN-Deconv}_{256} \to \text{LeakyReLU}_{0.2}$ 
 $\to \text{SN-Deconv}_{128} \to \text{LeakyReLU}_{0.2}$ 
 $\to \text{SN-Deconv}_{3} \to \text{Sigmoid}$ 
 $\to x \in \mathbb{R}^{32 \times 32 \times 3}$ |

Table B.5: Hyperparameters

| Dataset | Batch Size | $\lambda$ | $\eta$ | Learning Rate | Epochs |
|---------|-----------|-----------|--------|---------------|--------|
| Synthetic | 100 | $\{0.01, 0.003, 0.001,$ $0.0003, 0.0001\}$ | 1 | 0.0001 | 400 |
| MNISTS | 100 | $\{0.03, 0.01, 0.003,$ $0.001, 0.0003\}$ | 1 | 0.0001 | 400 |
| CIFAR-10 | 100 | $\{0.03, 0.01, 0.003,$ $0.001, 0.0003\}$ | 1 | 0.0001 | 1000 |
| CelebA | 100 | $\{0.03, 0.01, 0.003,$ $0.001, 0.0003\}$ | 1 | 0.0001 | 200 |

## B.3 ADDITIONAL EXPERIMENT RESULTS OF IMAGE DATASETS

### B.3.1 $L_2$ RECONSTRUCTION LOSS

$\lambda$ controls the trade-off between latent set compression and reconstruction. We can see in Fig. B.4 that as $\lambda$ goes up, the autoencoder's $L_2$ reconstruction loss increases (without pruning, marked by suffix "_non"), and eventually it converges to the reconstruction loss that is achieved after pruning all latent dimensions (marked by suffix "_all"). In this extreme case, the latent set approximately collapses into a single point and stores no useful information about the dataset. In terms of the four

regularizers' influence on the reconstruction, the ST regularizer seems to be the least sensitive to $\lambda$ compared with the other two that act directly on the latent STD vector $\boldsymbol{\sigma}$.

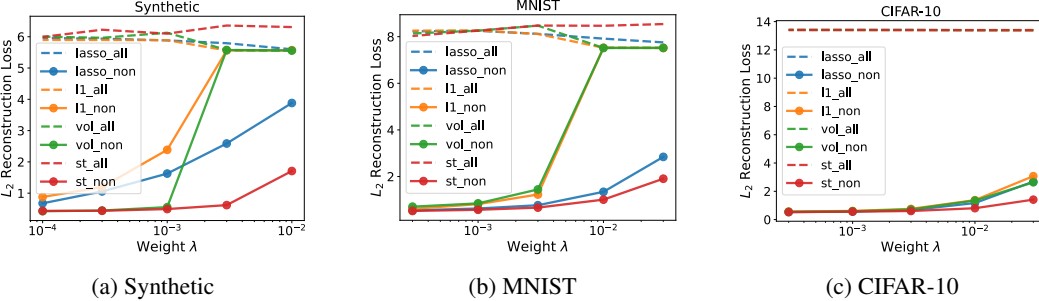

(a) Synthetic  (b) MNIST  (c) CIFAR-10

Figure B.4: $L_2$ Reconstruction vs $\lambda$

To better read Fig. 2, we need to determine which segments of the curves correspond to low quality reconstruction, so that we can ignore them. For this, it should be helpful to plot the autoencoders' reconstructions corresponding to different $L_2$ loss, so that we can grasp at what level of $L_2$ loss do the autoencoders' outputs have acceptable visual quality. In Fig. B.5 we can see that the image reconstructions have good quality when the $L_2$ reconstruction loss is under 2.

### B.3.2 Effect of $\eta$

As mentioned in §3.1, the supplemented volume penalty $\sqrt[m]{\prod_i \sigma_i + \eta}$ interpolates between $\sqrt[m]{\prod_i \sigma_i}$ and the $L_1$ penalty $\frac{1}{m}\|\boldsymbol{\sigma}\|_1$ gradient-wise. We investigate this by performing experiments at different $\eta$ in $\{0, 1, 3, 10, 30\}$ and plot their curves of Latent Set Dimensions vs $L_2$ Reconstruction Loss in Fig B.6. The hyperparameter configurations are listed in Table B.6. We can perceive this interpolation rather clearly in Fig. B.6b and B.6c that as $\eta$ increases, the dimension-reconstruction curve of the volume penalty converges to that of the $L_1$ penalty. The convergence to $L_1$ is not very apparent in Fig. B.6a probably because $\eta$ is much smaller than most STDs (See Table B.7).

Table B.6: Hyperparameters

| Dataset | Batch Size | $\lambda$ | $\eta$ | Learning Rate | Epochs |
|---------|-----------|-----------|--------|---------------|--------|
| Synthetic | 100 | $\{0.01, 0.003, 0.001,$ $0.0003, 0.0001\}$ | $\{0, 1, 3, 10, 30\}$ | 0.0001 | 400 |
| MNISTS | 100 | $\{0.03, 0.01, 0.003,$ $0.001, 0.0003\}$ | $\{0, 1, 3, 10, 30\}$ | 0.0001 | 400 |
| CIFAR-10 | 100 | $\{0.03, 0.01, 0.003,$ $0.001, 0.0003\}$ | $\{0, 1, 3, 10, 30\}$ | 0.0001 | 1000 |

However, $\eta = 0$ seems to be a special case whose curves differ greatly from the other $\eta$, and in Fig. B.6a and Fig. B.6c it is not as effective as the other $\eta$ in terms of dimension reduction. This is probably due to the gradient explosion induced by the scaling factor $\frac{\sqrt[m]{\prod_i \sigma_i}}{\sigma_i}$ for small $\sigma_i$.

### B.3.3 Ablation Study of Ordering Effect

It seems that even without Lipschitz regularization or latent regularization, the latent STD still in general correlates well with the explained reconstruction (or the degree of importance), as shown in Fig. B.7 and B.8. Although there is no clear answer to this, we hypothesize that this is because an unregularized decoder is still a Lipschitz function such that Theorem 3 still works. When minimizing the latent regularizers during training, although an unregularized decoder's Lipschitz constant is not upper bounded by spectral normalization, due to the gradient-based optimization it is not likely to increase abruptly, so practically we may regard the decoder's Lipschitz constant as being "softly"

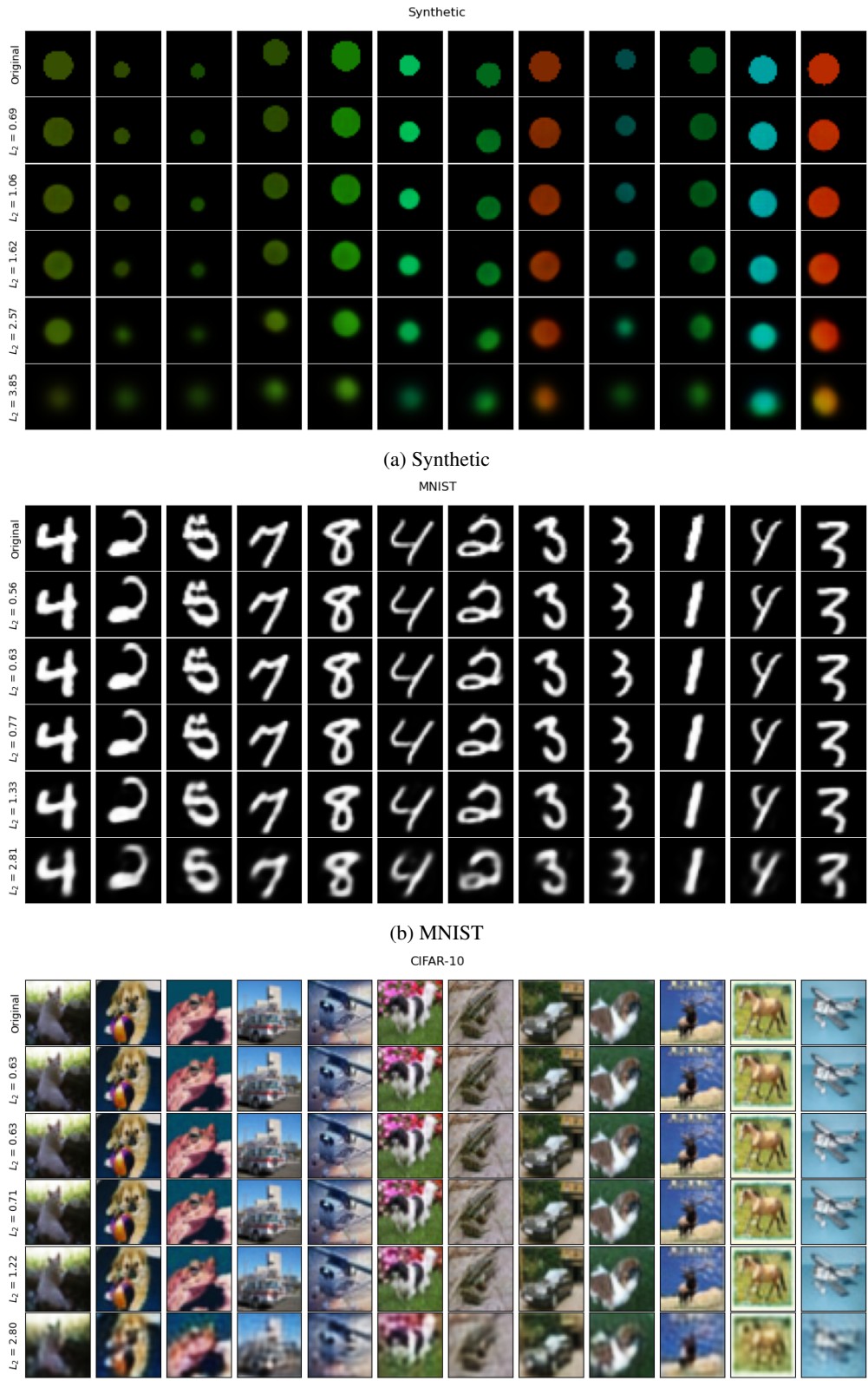

(a) Synthetic

(b) MNIST

(c) CIFAR-10

Figure B.5: $L_2$ Reconstruction Loss vs Sample

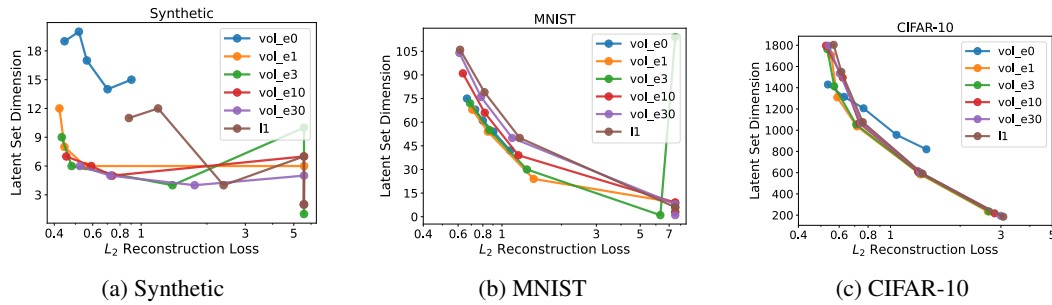

Figure B.6: Dimension-Reconstruction Curve vs $\eta$

upper bounded such that it still collaborates to some degree with the latent regularization. This hypothesis might explain the ordering effect in Fig. B.7.

However, in the case without latent regularization (Fig. B.8), both the autoencoders with and without Lipschitz regularization possess much more "rugged" explained reconstruction curves compared to Fig. 3 and B.7, which signifies that the latent dimensions with large STDs are more likely to correspond to trivial data dimensions when there is no latent regularization (Fig. B.8c is a good example of this ruggedness, especially when compared to Fig. B.7c and 3c). This agrees with our discussion in §3.4 about the latent regularization's role in producing the ordering effect.

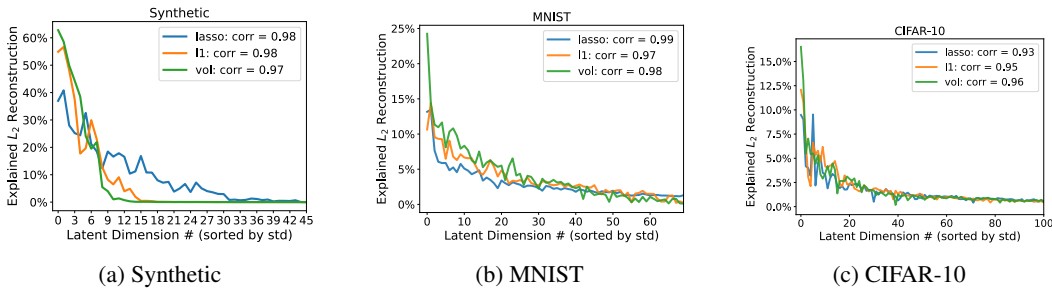

Figure B.7: (No Lipschitz Regularization) Explained Reconstruction vs Latent Dimension # (sorted in descending order of STD)

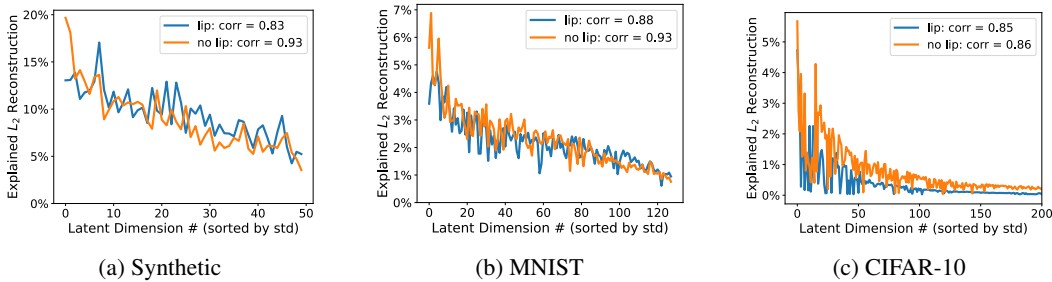

Figure B.8: (No Latent Regularization) Explained Reconstruction vs Latent Dimension # (sorted in descending order of STD)

### B.3.4    ABLATION STUDY OF ISOTROPIC SHRINKAGE

We list in Table B.7 the range of the principal latent dimensions' STDs for the cases with and without Lipschitz regularization on the decoder. Here "principal latent dimensions" refers to the latent dimensions left after cumulatively pruning those with the smallest STDs until their joint explained reconstruction exceeds 1%. We can see that the STDs in the cases without Lipschitz regularization

are orders of magnitude smaller than those with Lipschtiz regularization, and also smaller than their counterparts without latent regularization. This together with Fig. 4 shows that isotropic shrinkage happens when there is no Lipschitz regularization.

Table B.7: Ranges of $\sigma_i$

| Regularizer | Synthetic | | MNIST | | CIFAR-10 | |
|---|---|---|---|---|---|---|
| | Lip | No Lip | Lip | No Lip | Lip | No Lip |
| Lasso | 37.0 - 77.7 | 0.02 - 0.22 | 0.88 - 22.2 | 0.01 - 0.06 | 0.17 - 51.5 | 0.01 - 0.11 |
| $L_1$ | 46.7 - 102 | 0.01 - 0.44 | 3.12 - 29.1 | 0.01 - 0.05 | 0.20 - 50.7 | 0.004 - 0.10 |
| Vol | 105 - 1062 | 0.01 - 0.74 | 34.4 - 230 | 0.01 - 0.13 | 0.21 - 91.2 | 0.005 - 0.12 |
| No Reg | 2472 - 6048 | 4.81 - 11.4 | 81.9 - 171 | 0.91 - 2.41 | 29.3 - 882 | 0.10 - 3.17 |

### B.3.5 DOWNSTREAM KNN CLASSIFICATION TASKS

We perform KNN classification on MNIST and CIFAR-10 using the latent representation learned by different sparse AEs of similar sparsity levels (See Fig.B.9). In short, all methods are comparable on MNIST (Table B.8), but LV has much better performance on CIFAR-10 (Table B.9).

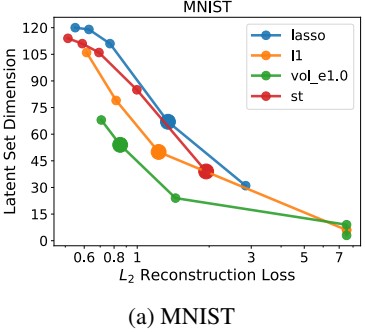
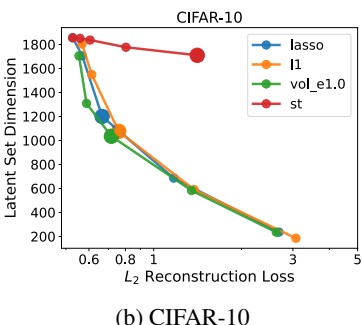

(a) MNIST

(b) CIFAR-10

Figure B.9: Cases Selected for Downstream Classification Tasks in Table 1 and 2 (Big dots mark the selections).

Table B.8: Prediction Scores of Latent KNN Classifiers (k=5) on MNIST

| Regularizer | Recall | Precision | Accuracy |
|---|---|---|---|
| Lasso | 97.19% ± 0.02% | 97.22% ± 0.02% | 97.22% ± 0.02% |
| $L_1$ | **97.29% ± 0.08%** | **97.33% ± 0.09%** | **97.32% ± 0.08%** |
| ST | 95.63% ± 0.20% | 95.70% ± 0.17% | 95.70% ± 0.19% |
| Vol | 96.68% ± 0.03% | 96.73% ± 0.03% | 96.73% ± 0.03% |
| Raw Data | 97.26% | 97.31% | 97.29% |

Table B.9: Prediction Scores of Latent KNN Classifiers (k=5) on CIFAR-10

| Regularizer | Recall | Precision | Accuracy |
|---|---|---|---|
| Lasso | 34.34% ± 0.23% | 41.97% ± 0.43% | 34.34% ± 0.23% |
| $L_1$ | 34.40% ± 0.09% | 41.93% ± 0.11% | 34.40% ± 0.09% |
| ST | 34.96% ± 1.57% | 37.64% ± 0.09% | 34.96% ± 1.57% |
| Vol | **41.08% ± 0.41%** | **42.15% ± 0.34%** | **41.08% ± 0.41%** |
| Raw Data | 33.98% | 43.04% | 33.98% |

# C  Tips for Hyperparameter Tuning, Model Training and Usage

## C.1  Hyperparameter Tuning

There are three hyperparameters—$K$, $\lambda$, $\eta$—to tune for least volume. However, usually we only need to take care of $\lambda$. Below we discuss them in detail.

1. $K$ is always first settled when constructing the decoder network and fixed afterwards, because the activation functions are fixed and the spectral normalization makes the linear layers 1-Lipschitz. Normally $K = 1$ works well, just like in all the experiments we did.

   However, if the data space is very high dimensional such that the data manifold has great length scale in terms of its geodesic distance, then this will in turn make the latent set large-scale after the flattening (check those large $\sigma_i$ in Table B.7 when the decoder is 1-Lipschitz). This might decelerate the training of the autoencoder, as the encoder then needs to dramatically increase its weights to scale up the latent code to reduce the reconstruction loss, which can be time-consuming. In this situation, increasing $K$ could reduce the length scale of the latent set and accelerate training.

   In case the length scale of the dataset really becomes a problem, to set $K$ and integrate it into the autoencoders:

   - As to the value of $K$, a practical choice is the dimension of the data space $X$ or any value of similar order of magnitude, provided the dataset is normalized. This value does not need to be exact because LV works for any $K$.
   - We can attach a scaling layer $f(x) = Kx$ to the input of $g$ to make it $K$-Lipschitz (*i.e.*, $d_{\text{new}} = d \circ f$).
     Alternatively, we can either attach $f$ to the output of the encoder $e$ (*i.e.*, $e_{\text{new}} = f \circ e$) or just increase the initial weights of $e$ to simply scale up the latent set. This keeps the decoder $g$ 1-Lipschitz.

2. Empirically, it is fine to leave $\eta$ at 1 by default for $K = 1$. In general, it had better be a value more than one or two orders of magnitude smaller than the largest latent STDs to make it not too $L_1$-like.

   However, it should be emphasized that in this work, this $\eta$ is introduced more like a concession, an unsatisfying alternative to the ideal but uneasy-to-implement *recursive* subspace extraction process introduced in §2.1. Without that recursive extraction and compression, the optimizer will keep compressing those trivialized latent dimensions unnecessarily even if they cannot be compressed any more. This could limit the complexity of the autoencoder thus prevent further flattening of the latent set, especially when the latent space is higher dimensional (given that for the same dataset, there are more trivial latent dimensions to compress in the latent space). This is the main drawback of this work in our opinion. A *reliable* realization of the recursive subspace extraction process is the missing piece that may unleash the full potential of Least Volume, thus is worth researching in the future.

3. $\lambda$ controls the trade-off between reconstruction and dimension reduction. This needs the user to set a reasonable threshold $\delta$ for the reconstruction error $\epsilon$, and tune $\lambda$ to make $\epsilon < \delta$. It is hard to tell what a proper $\delta$ is, given that different people have different tolerance to $\epsilon$ for different problems. The best practice for tackling this dilemma is probably transparency and openness: always report $\delta$ in the LV result.

   We can infer that when the autoencoder's latent dimension $m$ needs to be changed, $\lambda$ should be adjusted in proportion to $m$ to negate the scaling variation caused by the factor $1/m$ in Eqn. 5, so that the gradients induced by the volume penalty on the model's parameters have consistent magnitudes across different $m$.

## C.2  Model Training

At the start of each training, if $\lambda$ is too large, it is likely that the volume penalty will prevent the reduction of reconstruction error, trapping the model parameters at the local minima. In case this happens, it is practical to set a schedule for $\lambda$ that gradually increases it to the target value as the training proceeds. If $\lambda$ is set correctly, normally at the start of each training the reconstruction loss will immediately decrease, whereas the volume penalty will increase for a short period of time before

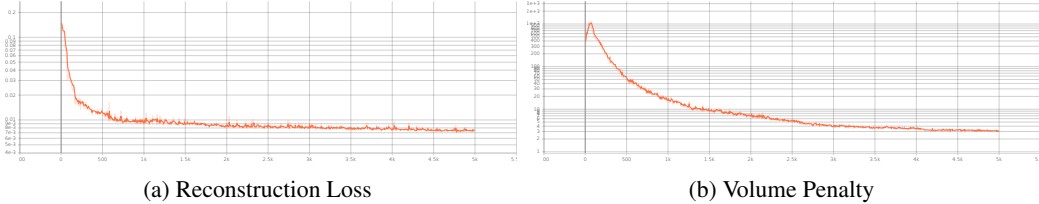

(a) Reconstruction Loss  (b) Volume Penalty

Figure C.10: Typical training history of LV.

it starts to drop, as shown in Fig. C.10. This is because when $K$ is not large, the latent set has to expand itself first to reduce the reconstruction loss, according to Theorem 3.

As to determining the number of epochs for training, a useful strategy is to inspect the history curve of the volume penalty, as usually the reconstruction loss converges faster than the volume penalty (Fig. C.10). It is ideal to stop training when the volume penalty value converges. Monitoring the histogram of latent STDs throughout training is another useful and informative way to assess the dimension reduction progress.

### C.3 MODEL USAGE

To quickly determine the compressed latent set's dimension after training LV models, rather than tediously evaluating the explained reconstructions one dimension by one dimension, a much more convenient way is to illustrate all latent STDs in descending order in a bar plot and put the y-axis on log scale. In every LV application we have done so far (including those outside this work), there is always a noticeable plummet at a given index in the log-scale bar plot (a real example is shown in Fig. C.11). All latent dimensions before this plummet can be regarded as corresponding to the principal dimensions of the dataset, considering their significantly larger STDs and empirically also much larger explained reconstructions.

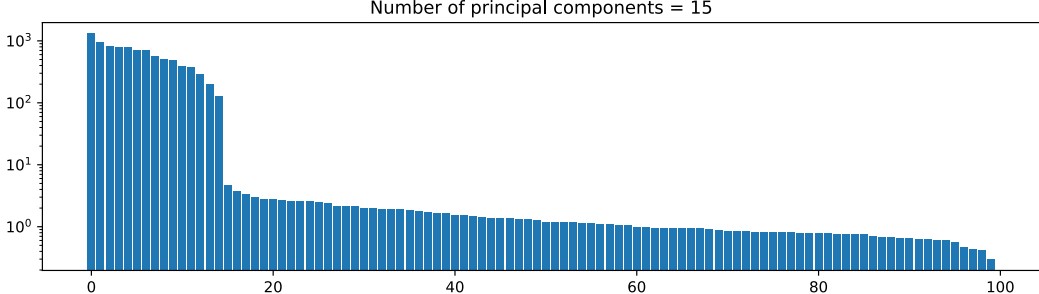

Figure C.11: Typical pattern in the latent STDs after LV regularization. In this example the plummet is at #15, which suggests the dataset might be 15-D or of lower dimension.

