# OpenReview forum: "Compressing Latent Space via Least Volume"
_ICLR.cc/2024/Conference — ICLR 2024 poster_

### Official Review · Reviewer_VztC · 2023-10-27

**Soundness:** 3 good
**Presentation:** 3 good
**Contribution:** 2 fair
**Rating:** 5
**Confidence:** 3

**Summary:**

This work proposes Least Volume (LV) a method that prunes latent dimensions of deterministic auto encoders by the combination of a specific regulariser in the latest space and specific constraints in the decoder. The regulariser is the product of the (empirical) variances of each dimension of the latent code; by encouraging minimisation of this term, the encoder is forced to “collapse” / “prune” its representations on some dimensions of the latent space. In order for these dimensions to be properly ignored by the decoder, the authors propose to employ a Lipschitz constraint, effectively preventing the decoder from being able to scale up the values of those dimensions. The authors provide several theoretical motivations for LV. Firstly, they discuss how their regulariser connects to volume and how, even though the autoencoder is deterministic, its continuity allows it to learn a topological embedding of the dataset. Secondly, they discuss how a bound on the reconstruction error by pruning dimensions with LV and how LV relates to PCA in the case of linear models. Finally, the authors also generalise the concept of the importance ordering of dimensions from PCA to the case of nonlinear models and LV, by relating the variance on each dimension to the reconstruction error on the output.

The authors perform some experiments on image tasks where they show how LV is able, for a fixed reconstruction penalty, to produce sparser latent spaces and furthermore perform ablation experiments to verify the importance of the components of LV, the Lipschitz constraint and the regulariser at the latent space.

**Strengths:**

- This work proposes a relatively simple idea that seems to work better than alternatives in practice.
- The geometric interpretation of the deterministic autoencoder is, as far as I am aware, novel.
- The regularizer proposed is, also as far as I am aware, novel.
- The theoretical discussions are a nice add-on and provide useful insights.

**Weaknesses:**

- Limited set of experiments; the authors mainly check reconstruction performance at a given sparsity level. As auto encoders are primarily used for their feature learning capabilities, it would be worthwhile to see how the features learned through LV fare in several downstream tasks.
- It would also be nice to compare against stochastic approaches, such as variational auto encoders, which can also have their own mechanisms of “pruning” latent dimensions, e.g., [1].
- The Lipschitz constraint on the decoder is not a novel contribution per se, as it has also been proposed at [2] via the means of a gradient penalty term. Furthermore, in light of this existing work, LV can be seen as employing a different regulariser in the latent space (compared to e.g., the squared L2 norm employed at [2])

[1] Automatic Relevance Determination for deep generative models, Karaletsos et al., 2015

[2] From variational to deterministic auto encoders, Ghosh et al., 2020

**Questions:**

Overall, I am a bit on the fence about this work. I like the theoretical discussion, but the paper lacks on the practical side and method-wise, given prior works such as [2], there is only the novelty of the regulariser in the latent space. My questions and suggestions are the following
- It seems that the only downstream task considered is a KNN classification where the results were a bit underwhelming. I would encourage the authors to do a more comprehensive evaluation of the feature learning capabilities of LV by expanding upon the classification task with linear probing (i.e., learning a classifier on top of the encoder) or other tasks considered in unsupervised models , such as semi-supervised learning, object detection and segmentation.
- In the experiment discussion, the authors mention that they use binary cross entropy for training, which deviates from their theory, but they perform evaluation with the L2 reconstruction error. Why is there this discrepancy? Furthermore, how is the binary cross-entropy used when you do not have binary data (e.g., CIFAR 10 and CelebA)?
- The authors argue that the CelebA results do not fit in the main text due to space constraints, however, there is almost half a page empty, so I would suggest that the authors use the empty space to move some more results in the main text.

---

> ### Author Response · Authors · 2023-11-16
>
> We appreciate your comments and suggestions. First of all, thanks for pointing out the nice work [2]. It was beyond our horizon and we agree that their Lipschitz constraint was introduced under a similar motivation, so we are happy to cite it in our revision. In the following, we answer your questions point-by-point.
>
> 1. Thanks for the suggestion. However, due to the time limit, we may not be able to include more comprehensive downstream tasks in our paper.
> 2. These images' pixel values are all normalized from [0,255] to [0,1], thus BCE can be employed as a reconstruction loss. BCE is not limited to the discrete binary classification case because gradient-wise for any label $y\in[0,1]$, we have $\frac{\partial BCE(p, y)}{\partial p} = 0 \Leftrightarrow p = y$, so minimizing it is equivalent to minimizing the other norm-based reconstruction losses like MSE and L2.
> We switched to BCE for the image autoencoder because it makes the AE converge much faster than MSE, probably because it assigns much higher gradient to wrongly reconstructed pixels.
> 3. Sure, we will certainly rearrange the paragraphs and figures in our revision to make it more consistent.

---

> > ### Comment · Reviewer_VztC · 2023-11-22
> > **Response to rebuttal**
> >
> > I would like to thank the authors for their response and I encourage them to revise their submission appropriately. Given that I believe that the main weakness is in the evaluation section and that still remains unaddressed, I will maintain my score.

---

### Official Review · Reviewer_8zx4 · 2023-10-30

**Soundness:** 3 good
**Presentation:** 2 fair
**Contribution:** 3 good
**Rating:** 8
**Confidence:** 4

**Summary:**

The authors introduce a method called "least volume" that regularizes an autoencoder to minimize the volume of the hypercube spanned by the standard deviations of its latent representations.

They directly minimize this product to via an additional loss term. The authors identify a necessary constraint for the latent representation volume to correspond to disentanglement, namely the K-Lipschitz constraint on the decoder network. Additionally the authors identify the possibility of exploding gradients under vanishing standard deviations and address this via an additional regularization parameter to their loss, which smoothly interpolates between minimizing volume and L1 regularization of the standard deviations. The authors prove that least volume is equivalent to PCA in the case of a linear autoencoder and show empirically that the disentanglement effect also persists for non-linear autoencoders. LV is compared to three other sparsity-inducing loss functions on the latent representation and shown to lead to the best reconstruction error when pruning latent dimensions. The appendix contains additional ablation studies and their interpretations.

**Strengths:**

Introduced Method:
- LV constitutes a simple and universal objective for disentanglement in autoencoders
- The authors show that latent dimensions are incentivised to tend towards exactly 0, which additionally benefits pruning
- The introduced objective operates in the metric of the latent space, which, if the latent space is structured well, can lead to improved soundness over methods that operate on data space

Writing:
- The authors explain their method in detail and provide many illustrating examples


Experiments:
- Extensive ablation studies are performed in the appendix

**Weaknesses:**

Introduced method:
- LV requires the autoencoder to be K-Lipschitz, which can be a large constraint in some cases
- Although the method is novel, the idea of minimizing the volume of the learned latent representations is not, see [PCA-AE](https://link.springer.com/article/10.1007/s10851-022-01077-z), [IRMAE](https://ar5iv.labs.arxiv.org/html/2010.00679). LV significantly differs from the mentioned papers, as it directly acts on the latent codes and does not require multistage training.Nonetheless, they should be mentioned.
- The authors mention [Sparse Feature Learning for Deep Belief Networks](https://papers.nips.cc/paper_files/paper/2007/hash/c60d060b946d6dd6145dcbad5c4ccf6f-Abstract.html), but do little to point out the similarities/differences to their method. The regularization term of the latent codes in that work is
$L_\text{vol} = \sum\limits_{i=1}^{m} \log(1 + h^2(z_i))$,
where $h$ is defined in Eq. 6 of the paper (notation was adjusted to fit the notation of the paper under review).
When we regard $h$ as a homeomorphism that just produces new latent codes and center those new latent codes to 0 we get
$\sqrt[m]{\exp(L_\text{vol})} = \sqrt[m]{\prod\limits_{i=1}^{m} (\sigma_i + 1)}$.
This is equivalent to LV with $\eta = 1$.
If the authors can point out what their contribution with regards to that already published regularization term is, it would in my opinion severely strengthen the case for this work.




Writing:
- In the related work section the authors describe the invariance of their method to translation of the latent codes. However in the next sentence it is stated that: "This equivalent latent set is then one that has zero-STDs along many latent dimensions, which is what this work tries to construct". It is not clear whether this refers to the translated latent set of LV or of aforementioned methods. In case it refers to LV, the zero-STDs are present also before translation (so it has *still* zero-STDs not only *then*). In case it refers to other methods the authors need to elaborate more on the claim that established methods already produce close to zero-STD latent codes in many latent-dimensions and how their method then differs from those established methods apart from allowing for translational invariance, which would not be a strong contribution by itself. In my opinion this would also generally be a good addition to the related works section.
- In section B.3.3 in the appendix the authors claim that latent STD still correlates well with explained reconstruction if no Lipschitz constraint is applied because the decoder is naturally a Lipschitz function. While this is correct the Lipschitz constant of the decoder can be arbitrarily increased. It is not clear whether the argument of small updates of gradient-based optimization preserving the Lipschitz constant holds after cumulative gradient updates and sufficient training time. The observed effect might well be due to specific training dynamics and model initialization and, while it is interesting to report, a clear conclusion is hard to draw.

Experiments:
- The autoencoders are trained with BCE loss on image datasets, but are evaluated in terms of L2 reconstruction. I do not see the justification for evaluating the trained models on a different metric than the training target. If there is a reason behind this the authors should clarify it.

In my opinion, the main weakness of this work is that there is not sufficient methodological comparision to established work in the field to separate the novel concepts from existing ones. If this is amended I would be willing to significantly change my rating in favor of acceptance.

### A few minor suggestions:

Introduction
- "This openness means for a conditional generative model trained in such Z, if its one prediction ẑ is not far away from its target
z ∈ U , then ẑ will also fall inside U ⊆ Z, thus still be mapped back into X by g rather than falling outside it." $\rightarrow$ this is only true for g with small Lipschitz constants
- "Additionally, people also hope the latent representation can indicate the amount of information each dimension has,
so that the data variations along the principal dimensions can be easily studied, and trade-offs can be
easily made when it is necessary to strip off less informant latent dimensions due to computational cost" $\rightarrow$ This is a good place for some citations

Methodology and Intuition
- Figure 1: The arrows are labelled with "-v". What does that mean? Is it another abbreviation for LV?
- Section 2.3: "...each homeomorphic latent set h(Z) with h ∈ H must also be an optimal solution to J, thus residing in Z⋆" . $\rightarrow$ How does this combine with the Lipschitz constraint on $g \circ h^{-1}$?

Theoretical Analysis
- Section 3.2: "Although some regard an errorless autoencoder as “trivial” since it only learns an one-to-one representation..." $\rightarrow$ citation needed
- Section 3.2: "Thus if X is a smooth manifold, then obtaining the least dimensional Z through a smooth autoencoder can provide a lower bound of X ’s dimensionality." $\rightarrow$ upper bound
- Section 3.4: In Proposition 2, define $\lambda$

Experiments:
- Figure 2: It might be more instructive if the axes of the figures were swapped. Also a more thorough explaination of the figure and the meaning of the markers in the caption would be nice.

Appendix
- Section A.4: "Since S ⋆ —the volume-minimized S— is diagonal..." $\rightarrow$ refer to theorem 1 here
- Section B.2.4: Listing the hyperparameter grid for different experiments together is confusing. Separate the experiments where you vary for $\lambda$ and $\eta$ visually. Otherwise the sentence: "By default η = 1 for all experiments" stands in direct conflict with Table B.5.
- Section B.3.2: Missing hyperref in the beginning

**Questions:**

s.a.

---

> ### Author Response · Authors · 2023-11-16
>
> Thanks for your meticulous comments and suggestions. In the following, we give our detailed response to your questions point-by-point.
>
> ## Introduced method
>
> Thanks for mentioning these great works. We actually cited IRMAE in our paper, and we will add PCA-AE in our revision.
>
> The student-t regularizer is not equivalent to the volume regularizer. First of all, for a mini-batch based training involving the student-t regularizer, the average over the batch samples is taken outside the regularizer term, i.e., $\frac{1}{n}\sum_j^n \sum_i^m \log(1+h^2(z_i^{(j)})) \neq \sum_i^m \log(1+ \frac{1}{n}\sum_j^nh^2(z_i^{(j)}))$. Second, even if the average is taken inside the term, the equation should instead be $\sqrt[m]{\exp(L_{\text{vol}})} = \sqrt[m]{\prod_i^m (\sigma_i^2+1)}$. In other words, the student-t regularizor is like minimizing the square of the volume, i.e., taking the variance rather than the STD as the side length of the cuboid. Just like the difference between L2 and L1, this makes it more inclined to reduce larger $\sigma_i$, which leads to the discrepancy between “vol” and “st” in Fig. 2.
>
> ## Writing
>
> 1. Sorry for the confusion. It refers to the translated latent set of LV. What we are trying to convey here is that **in terms of the latent representation’s sparsity,** **the *absolute value* of a latent code dimension $z_i$ actually does not matter**, as what really matters is its STD $\sigma_i$. For a given latent dimension $z_i$, if it stays at 0 across all the data samples (thus making the latent code vector sparse in the traditional sense and leading to some improvement in downstream tasks), then the latent code vector should be equally sparse and good if this $z_i$ stays at any constant $c$ other than 0, because it can be easily made back to 0 with a simple translation that any complex neural network can learn. In other words, the sparsity of a latent code $\mathbf{z}$ should not be determined by how many $z_i = 0$ it has, but by how many $\sigma_i = 0$ it has.
> Our LV method is focused on the latter generalized definition of sparsity, and encourages sparsity this way. A sparse latent dimension obtained by LV does not necessarily have its mean equal to 0, but can be easily turned into a traditionally sparse dimension with a simple translation.
> 2. Yes, we are just providing our hypothesis about this phenomenon. We would like to clarify it more in our revision in case it induces confusion.
>
> ## Experiments
>
> These images' pixel values are all normalized from [0,255] to [0,1], thus BCE can be employed as a reconstruction loss. BCE is not limited to the discrete binary classification case because gradient-wise for any label $y\in[0,1]$, we have $\frac{\partial BCE(p, y)}{\partial p} = 0 \Leftrightarrow p = y$, so minimizing it is equivalent to minimizing the other norm-based reconstruction losses like MSE and L2.
> The L2 loss is more like a metric in our theoretical framework for quantifying the accuracy of reconstruction. Just like we usually use CE as the loss function for classification problems, but we discuss the classification performance in terms of precision, recall, etc.
> We chose BCE as the loss function for image reconstruction because it leads to much faster convergence, reducing L2 error much faster than L2 itself and MSE.
>
> Thanks for letting us know the main weakness of our paper to make it better. Sadly due to time limit we may not be able to included a comprehensive set of experiments.
>
> ## Minor Suggestions
>
> We are happy to streamline our paper according to your suggestion, and thank you again for providing such as detailed feedback.

---

> > ### Comment · Reviewer_8zx4 · 2023-11-21
> >
> > Thank you for your detailed response. Below are some follow-ups on the comments made.
> > ## Student-t Regularizer
> > The delineation between the introduced method and the student-t regularizer is now much more clear to me and shows the uniqueness and advantages of LV much more prominently.
> > ## Writing
> > Thank you for clarifying, I fully agree with the statements made.
> > ## Experiments
> > As you stated, training the model with BCE induces different training dynamics than training with reconstruction loss. It is common to employ a different evaluation metric during validation, as illustrated by your example for classification tasks. However, when one chooses to use a different evaluation metric, it is usually justified by that metric capturing some desirable aspect of the data better. My question was, whether there was such a desirable aspect that the L2 reconstruction captures better than the BCE loss in this case? Since all the methods benchmarked act on the latent space, does the change in evaluation metric leave their relative performance to each other unaffected?
> >
> > Overall, the novelty of this concept has been made more obvious to me and I will adjust my rating.

---

> ### Author Response · Authors · 2023-11-21
>
> That is a good question. The reason why we prefer L2 distance over the other metric for evaluating the reconstruction quality is as follows:
>
> 1. The Lipschitz constant of a function is usually defined w.r.t. the L2 distance in the domain and codomain, so using the L2 distance to evaluate the  distance between images can provide us a consistent and intuitive sense of distance. In addition, Theorem 3, Proposition 1 and the spectral normalization we used are based on the usual L2 definition of Lipschitz constant, so L2 distance is chosen by default in this work as the metric for consistency.
> 2. That being said, it does not mean theoretically we must always stick with the L2 distance. We may find a another better distance for measuring the discrepancy between images (e.g., the Riemannian distance on the image manifold—although it is challenging to retrieve, or the “perceptual similarity” learned by a VGG network [1]—although it might be a pseudodistance), and redefine the Lipschitz constant accordingly to provide a better sense of the data variation (e.g., using the L2 distance in the latent space Z and the Riemannian distance in the image data space X). However, this may in consequence require us to overhaul several theorems we established, and some of them may be destroyed. For instance, our conclusion about $\Sigma$ in Proposition 1’s proof does not hold any more, as the spectral radius of a matrix is defined in term of L2.
> 3. BCE is not a distance function, because it does not have the symmetric property a distance function needs to have. This might introduce some complexity into our analysis, so we didn’t use it as the metric.
>
> [1] Zhang, Richard, et al. "The unreasonable effectiveness of deep features as a perceptual metric." *Proceedings of the IEEE conference on computer vision and pattern recognition*. 2018.
>
> We hope this answers your question. In addition, we revised the paper according to your suggestions. Mainly, we added Table 1 in the literature review section to illustrate the difference of LV against the other methods. Thanks a lot for increasing the score.

---

### Official Review · Reviewer_nEBV · 2023-10-31

**Soundness:** 4 excellent
**Presentation:** 3 good
**Contribution:** 3 good
**Rating:** 6
**Confidence:** 4

**Summary:**

The paper proposes a regularization scheme for autoencoders that allows for automatic determination of the latent dimension. This is achieved by measuring the product of axis-aligned standard deviations of the latent codes. This is simple and seems to work well for its purpose. The proposed regularizer is studied extensively theoretically which shows links to the ordering of latent dimensions known from PCA.

**Strengths:**

* The presented regularizer is simple, intuitive, and seems easy to implement (this is very good).
* The proposed approach is well-studied theoretically
* Experimentally, the approach appears to work well.

**Weaknesses:**

* To me, it seems like the main weakness is that the approach relies on several tuning parameters ($K$, $\eta$, $\lambda$) and it is unclear how to set these and how they interact with each other.
* The experiments are somewhat limiting. They basically show that the approach can succeed in reducing latent dimension while retaining good reconstruction. It would have been nice with experiments showing that the resulting models then were more useful for some task.
* I miss experiments on the interaction between tuning parameters.
* The approximation $L = J + \lambda L_{vol}$ should be more prominent (it's easily missed).

**Questions:**

* How do you tune the free parameters?
* On page 5, what is $\pi$? I couldn't figure it out from the context.
* The error bars in fig 2 are quite small. Does that imply that things are stable wrt choice of $\lambda$ or that the studied $\lambda$-range was too small?

---

> ### Author Response · Authors · 2023-11-16
>
> We sincerely appreciate your comments and suggestions. In the following, we answer your questions point-by-point.
>
> 1. We did some sensitivity studies regarding $K, \eta, \lambda$ in Appendix B. In summary:
>     1. $K$ is always first settled when constructing the decoder network and fixed afterwards, because the activation functions are fixed and the spectral normalization makes the linear layers 1-Lipschitz.
>     Its absolute value won’t affect LV’s dimension reduction ability, because if you can find a least-volume latent set $\mathcal{Z}$ with an encoder $e$ and a 1-Lipschitz decoder $g$, then for a new encoder $e'$ and a new $r$-Lipschitz decoder $g'$, you must be able to find a new least-volume $\mathcal{Z}'$ that is similar to $\mathcal{Z}$ with a scale factor of $1/r$ (i.e., $\mathcal{Z}' = \mathcal{Z}/r$), simply by making $g' = g \circ h$ and $e'= h^{-1}\circ e$, where $h(z') = r\cdot z'$.
>     In short, as long as it is bounded, you don’t need to worry about its value.
>     2. $\eta$ is studied in Sec. B.3.2 and Fig. B.6, which shows it indeed interpolates between $\sqrt[m]{\prod_i\sigma_i}$ and $\frac{1}{m}\|\sigma\|_1$. In practice, we almost always set it to 1, not only because it worked well enough in all cases we encountered, but also helps avoid gradient issue by lower bounding the denominator in Eqn. 4.
>     $K$ should have some interaction with $\eta$, because $K$ may scale up or down the latent set and thus scale $\sigma_i$, which changes the ratio between $\sigma_i$ and $\eta$.
>     3. $\lambda$ is set after $K$ and $\eta$ are fixed. $K$ scales the latent set and may $K$scales its STDs $\sigma_i$$K$$\lambda$ much, because 1) $K$ and $\eta$ are fixed and we don’t adjust them frequently. 2) The scaling of $\sigma_i$ and $\eta$ should be cancelled out in Eqn. 4.
> 2. $\pi: Z'\times c\to Z'$ is the projection map that strips off the dimensions correspond to $c$ and only keeps those correspond to $Z'$, i.e., removing all the latent dimensions that have constant values. As a toy example, $\pi: X_1\times X_2 \times X_3\to X_1\times X_2$ means for the vector $x = [4,5,6]$, we get $\pi(x)=[4,5]$. A quick reference can be found here: [https://en.wikipedia.org/wiki/Projection_(set_theory)](https://en.wikipedia.org/wiki/Projection_(set_theory)?oldformat=true)
> 3. Yes, this means it is stable w.r.t. $\lambda$. We don’t think the range of $\lambda$ is small, because as you can see in Fig. B.4 in Sec. B.3.1, the largest $\lambda$ in each case is large enough to distort the reconstruction significantly.
>
> In addition, we will address some of the minor issues you mentioned in our revision.

---

> > ### Comment · Reviewer_nEBV · 2023-11-16
> > **Thanks**
> >
> > I appreciate the detailed feedback on the parameter tuning. Thanks.

---

### Official Review · Reviewer_k8im · 2023-11-06

**Soundness:** 3 good
**Presentation:** 3 good
**Contribution:** 2 fair
**Rating:** 5
**Confidence:** 4

**Summary:**

This paper proposes the least volume regularization method for autoencoder training. It reduces the geometric mean of standard deviation of latent vectors, while imposing Lipschitz continuity on the decoder to prevent trivial solutions. By applying this regularizer, latent representations become sparser, effectively reducing latent dimensionality while preserving reconstruction capabilities. It is shown that employing the least volume regularization in linear autoencoder results in PCA. Numerical experiments are performed to compare the least volume regularization with other sparsity-inducing methods.

**Strengths:**

- The mathematical derivations are sound, and minimizing volume while setting an upper bound on the decoder’s Lipschitz constant seems a clever approach.
- The connection established between PCA and the least volume regularization for linear autoencoders is intriguing and serves as a compelling rationale for this method.
- The notion of “explained reconstruction” seems convincing for investigating the importance of each latent dimension in the autoencoder.

**Weaknesses:**

- The practical applicability of the proposed regularization method and the usefulness of the obtained low-dimensional latent spaces seem quite limited. The appendix contains some experiments of k-NN prediction in the latent space, but the accuracy is much lower than state-of-the-art methods.
- The analysis of latent dimensions is primarily centered around reconstruction error, but the paper could explore other ways to assess the properties of obtained latent dimensions. For example, investigating the impact of small variations along each dimension on images would be insightful.
- Certain algorithmic details, such as the hyperparameters for the power method in spectral normalization of decoders, are missing.

**Questions:**

- If the data sets are composed of distinct clusters, would this least volume regularization provide meaningful cluster structures?
- Can this method be used to estimate the intrinsic dimensionality of a data set? How are the obtained latent set dimensions close to the exact data set dimensions (if known)?
- How do the overall outcomes, such as latent set dimensions, reconstruction losses, and the correlation between explained reconstruction and the standard deviations of latent dimensions, change when the dimension of the encoding latent space is altered?
- Is the regularizer computed on minibatch? If so, how is the minibatch estimate reliable? How is this method scalable?
- What is the meaning of ‘Raw Data’ in Tables B.7 and 8? Does it represent the performance without regularization? Moreover, why does volume regularization show distinct performance compared to other methods in CIFAR-10 experiments, even if they have similar latent set dimensions and similar reconstruction loss?
- Is this regularization method only effective in the autoencoder setting? How can this regularization method be extended to other applications?

[Minor comments]
- $\mathcal{H}$ is not defined in Section 2.3.
- There is a discrepancy between the loss function described in Section 5.1 (binary cross-entropy) and the loss types shown in the figures (L2 reconstruction loss).
- On page 19, Table B.8. -> Table B.6.

---

> ### Author Response · Authors · 2023-11-16
>
> We sincerely appreciate your comments and suggestions. In the following, we give our detailed response to your questions point-by-point.
>
> 1. Yes. This is due to the homeomorphism that a well-trained autoencoder can establish between the dataset and the latent set (Theorem 2). If the dataset consists of disconnected components, then because the connectedness of a set is invariant under homeomorphism, these components will remain disconnected and form clusters in the latent space. The least volume regularization is just searching for a lower dimensional latent set among all the homeomorphic latent sets that an autoencoder can establish, without destroying that homeomorphism.
> 2. This depends on your definition of intrinsic dimensionality (ID). If you assume the dataset structure to be *locally Euclidean* of certain dimension and define the ID as that *local* dimension, then least volume cannot always retrieve that local ID, for instance in the following complex scenarios :
>     1. The dataset is the union of several manifolds of different local IDs.
>     2. The dataset is a manifold that cannot be embedded in a Euclidean space of the same dimension, e.g., sphere, Klein bottle.
>     3. The dataset is not a manifold, at least not locally Euclidean somewhere, e.g., the union of several *intersecting* manifolds.
>     4. The data manifold forms a non-trivial knot in the data space.
>
>     In these scenarios, traditional methods based on neighborhood graph (e.g. isomap) are probably better choices, though they may also have difficulty handling them. However, this doesn’t mean least volume is worthless, because what it tries to retrieve is the *global* ID, i.e., the dimensionality of the lowest dimensional Euclidean space that the whole dataset can be topologically embedded in, whereas the local methods are not designed to obtain this. Thus, LV is a *global, bidirectional* complement to those *local* methods. We found it generally works well in retrieving the global ID.
>
> 3. We didn’t tested this meticulously, but we didn’t experience any noticeable difference in the result when we double the latent dimensions.
> 4. Yes, it is evaluated on minibatch. We found that in general a larger batch size (≥64) helps stabilize the training more than a smaller one (≤32), probably because it provides a better estimation of the STD. So we set the batch size to 100. However, typical small batch size like 32 still works.
> 5. Raw Data means the KNN is performed in the data space without any encoding. We also wonder why it has much better performance on CIFAR-10. We hypothesize that it flattens the dataset more than the other methods in the latent space (as Figure. 2c shows), such that the Euclidean distance in the latent space agrees better with the geodesic distance that is intrinsic on the CIFAR-10 manifold, which makes KNN result better.
> 6. It is not restricted to autoencoders. For instance, we can replace the decoder with a regressor and replace the reconstruction loss with the regression loss. In this case, the latent set is still forced to preserve information and thus won’t be compressed trivially. However, such use cases are theoretically more complicated than the autoencoding case and deserves a separate work. Specifically, for the current autoencoding case, we can readily resort to the theories about homeomorphism in general topology. For the general use case, though, we think it pertains to quotient map and quotient topology, which are more intricate topics. For now, to extend LV to the other applications, we can disengage the encoder and decoder and use them separately for different tasks, such as clustering and data generation.
>
> ## Minor Comments
>
> 1. $\mathcal{H}$ is the “a given set of $h$”, i.e., the set of $h$ that $g$ and $e$ have enough complexity to represent. We will rephrase it to make it clearer.
> 2. These images' pixel values are all normalized from [0,255] to [0,1], thus BCE can be employed as a reconstruction loss. BCE is not limited to the discrete binary classification case because gradient-wise for any label $y\in[0,1]$, we have $\frac{\partial BCE(p, y)}{\partial p} = 0 \Leftrightarrow p = y$, so minimizing it is equivalent to minimizing the other norm-based reconstruction losses like MSE and L2.
> 3. Thanks for pointing this out. We will correct it.

---

> > ### Comment · Reviewer_k8im · 2023-11-22
> >
> > Thank you for the detailed point-by-point answers to the questions. According to the answers, I could appreciate the idea of least volume regularization more.
> > I'll keep my score because the first point in the weaknesses, which is my main concern about the paper, still remains. However, I'm okay with it if the paper gets accepted.

---

### Meta-Review · Area_Chair_KY5Q · 2023-12-12

**Metareview:**

This paper was borderline, but I'm inclined to accept it. I think it proposes an interesting form of regularization and develops it in significant detail, even if the experimental results could be better. The authors also did a good job in answering the reviewers' questions.

**Justification For Why Not Higher Score:**

Reviewers not that enthusiastic

**Justification For Why Not Lower Score:**

See metareview.

---

### Decision · Program_Chairs · 2024-01-16

Accept (poster)